# Decreased Brain Structural Network Connectivity in Patients with Mild Cognitive Impairment: A Novel Fractal Dimension Analysis

**DOI:** 10.3390/brainsci13010093

**Published:** 2023-01-03

**Authors:** Chi Ieong Lau, Jiann-Horng Yeh, Yuh-Feng Tsai, Chen-Yu Hsiao, Yu-Te Wu, Chi-Wen Jao

**Affiliations:** 1Institute of Biophotonics, National Yang Ming Chiao Tung University, Taipei 112, Taiwan; 2School of Medicine, College of Medicine, Fu Jen Catholic University, New Taipei 242, Taiwan; 3Dementia Center, Department of Neurology, Shin Kong Wu Ho-Su Memorial Hospital, Taipei 111, Taiwan; 4Applied Cognitive Neuroscience Group, Institute of Cognitive Neuroscience, University College London, London WC1N 3AZ, UK; 5Department of Neurology, University Hospital, Taipa 999078, Macau; 6Department of Neurology, Shin Kong Wu Ho-Su Memorial Hospital, Taipei 111, Taiwan; 7Department of Diagnostic Radiology, Shin Kong Wu Ho Su Memorial Hospital, Taipei 111, Taiwan; 8Brain Research Center, National Yang Ming Chiao Tung University, Taipei 112, Taiwan; 9Department of Research, Shin Kong Wu Ho-Su Memorial Hospital, Taipei 111, Taiwan

**Keywords:** MCI, cognitive function, fractal dimension, brain structural network, Alzheimer’s disease

## Abstract

Mild cognitive impairment (MCI) is widely regarded to be the intermediate stage to Alzheimer’s disease. Cerebral morphological alteration in cortical subregions can provide an accurate predictor for early recognition of MCI. Thirty patients with MCI and thirty healthy control subjects participated in this study. The Desikan–Killiany cortical atlas was applied to segment participants’ cerebral cortex into 68 subregions. A complexity measure termed fractal dimension (FD) was applied to assess morphological changes in cortical subregions of participants. The MCI group revealed significantly decreased FD values in the bilateral temporal lobes, right parietal lobe including the medial temporal, fusiform, para hippocampal, and also the orbitofrontal lobes. We further proposed a novel FD-based brain structural network to compare network parameters, including intra- and inter-lobular connectivity between groups. The control group had five modules, and the MCI group had six modules in their brain networks. The MCI group demonstrated shrinkage of modular sizes with fewer components integrated, and significantly decreased global modularity in the brain network. The MCI group had lower intra- and inter-lobular connectivity in all lobes. Between cerebral lobes, the MCI patients may maintain nodal connections between both hemispheres to reduce connectivity loss in the lateral hemispheres. The method and results presented in this study could be a suitable tool for early detection of MCI.

## 1. Introduction

Alzheimer’s disease (AD) is a chronic neurodegenerative disease and is the most common form of dementia. Mild cognitive impairment (MCI) is widely considered to be an intermediate stage of cognitive impairment between normal cognitive aging and dementia-related changes [1]. In clinical practice, MCI is defined based on a greater degree of cognitive decline than would be expected at any given age [2]. Patients with MCI have a high rate of progression to dementia in a relatively short period of time [3]. Numerous studies have shown that MCI can be considered to be a prodromal stage of AD. Identification and treatment of MCI is very important because this disease progresses to dementia at a rate of between 8% and 15% per year [4]. For the diagnosis of MCI, clinical assessment is complicated by the heterogeneity of cognitive reserve and the diversity of daily functioning [2]. In addition to cognitive decline, MCI patients show cortical atrophy in some specific regions. As compared with the brains of healthy elderly people, the cortical thickness of MCI patients is significantly reduced mainly in the medial temporal regions and in some areas of the frontal and parietal cortices [5]. As the disease progresses from MCI to AD, more pronounced cortical thinning is found in the lateral temporal lobe and is more pronounced in the left hemisphere [5]. Other volumetric studies of MCI have reported volume loss in other parts of the brain in addition to the hippocampus and medial cortex, including the para hippocampus, amygdala and fusiform gyrus, lateral temporal lobe, cingulate gyrus, insula, parietal lobe, frontal lobe, and occipital lobe [6,7,8].

Cortical sulci can be regarded to be complex natural structures that may be difficult to quantify. Fractal dimension (FD) was introduced by Benoit Mandelbrot in 1967 [9] as a quantitative parameter to describe the morphological changes of complex objects [10,11,12]. FD descriptors can provide quantitative information related to cortical convolution, and changes in FD values are said to indicate cortical abnormalities [12]. For quantifying cerebral morphological complexity, the FD analysis method is superior to traditional volumetric methods because it exhibits less variability and smaller gender effects [12]. Therefore, FD has been widely used to quantify shape complexity and morphological changes in many neurodegenerative diseases [10,11,12,13]. Furthermore, FD has been shown to be a promising tool with good sensitivity in capturing atrophy processes [14]. In the early stages of AD, cortical FD can be used as a biomarker to detect structural changes in neurodegenerative diseases [15].

A structural network analysis of the brain can provide rich quantitative insights into the organization, development, and function of complex brain networks [16]. By assessing the properties of modules in brain networks, connections and relationships between brain network structures and functions can be explored [17]. For neurodegenerative diseases such as dementia [18], multiple sclerosis [19], and AD [20], brain networks based on cortical features can show anatomical connections between specific brain regions [21,22,23]. Cognitive assessment predominates the diagnosis of MCI, but reliable estimates of structural changes in specific brain regions are still lacking [2]. Furthermore, intra- and inter-lobular connectivity is an important property of a brain network analysis [24]. In neurodegenerative diseases, monitoring structural changes in the brain may be useful to understand pathophysiology and to prevent or modify progressive neurodegeneration. To date, FD measurements of specific brain regions in MCI patients and whether these atrophies lead to changes in brain structural network connectivity have not been well explored.

In this study, we investigated and compared the brain structural network connections in healthy and MCI groups. First, the brain was parcellated into 68 subregions as relative focal regions using the Desikan–Killiany atlas [25]. The FD values of the parcellated regions in both groups were measured and compared to the assessment of significantly atrophied regions. Then, we calculated correlations of FD values between paired brain regions, generating 68 × 68 correlation matrices for both groups. These correlation matrices were then used for modularity analysis to build a structural connectivity network and to calculate intra- and inter-modular connections throughout the brain and its lobes. Network properties, such as the modular numbers, modularity, and intra- and inter-lobular connectivity of each group, were also measured and compared. We hypothesized that changes in structural network properties and changes in intra- and inter-lobular connections may occur between healthy control participants and MCI patients.

## 2. Materials and Methods

### 2.1. Participants

Table 1 summarizes the demographics and characteristics of control participants and the MCI group in this study. There wa a total of 30 patients with MCI (male/female, 14/16) and 30 (male/female, 15/15) healthy control participants in this study. This study was conducted according to the guidelines of the Declaration of Helsinki and was approved by the Institutional Review Board of the Shin Kong Wu Ho-Su Memorial Hospital (IRB Number:20200104R). All of the participants were recruited from the Department of Neurology at the Shin Kong Wu Ho Su memorial Hospital. Sociodemographic variables such as age, sex, body weight, height and body mass index, years of education, and handedness were obtained from interviews and medical records. The inclusion criteria were as follows: (1) subjects aged 65 years and over; (2) a diagnosis of MCI according to Petersen’s criteria [26]; (3) a global rating of 0.5 on the Clinical Dementia Rating (CDR) scale; and (4) able to walk more than 10 m without walking aids. The exclusion criteria included (1) dementia; (2) brain tumors; (3) previous cerebral infarction or hemorrhage; (4) other known neurodegenerative or neuropsychiatric conditions; (5) the presence of an unstable orthopedic disease interfering with participation in the study; and (6) an education level of less than 6 years (elementary school). All participants including MCI patients and control participants were diagnosed by clinicians. All control group participants were free of central nervous system disorders and did not present any neurological abnormalities during the study period.

### 2.2. Image Acquisition and Cortical Feature-Based Structural Network

A 3T MRI scanner (Siemens, Erlangen, Germany) was applied to scan axial MRIs of the participants’ brains, including the whole brain and cerebellum. The parameters of the circular head coil for which T1-weighted images were acquired were: repetition time, 14.4 ms; echo time, 5.5 ms; matrix size, 256 × 256; 1.5 mm axial slice; field of view, 256 × 256 mm, and voxel size, 1.0 × 1.0 × 1.5 mm^3^. Each structural MRI dataset was normalized using DiffeoMap to presegmented and validated volume templates, and then applied the SPM8 toolbox to segment each normalized image into grey matter, white matter, and cerebrospinal fluid. Finally, each voxel of grey matter was anatomically aligned to 68 automatic anatomical landmark (AAL) structures by using the IBASPM (Individual Brain Mapping using Statistical Parametric Mapping) toolbox in MATLAB R2013b software (Mathworks, Natick, MA, USA). Then, the cortex was parcellated and aligned into 68 subregions of interest (ROIs) with the Desikan–Killiany cortical atlas (DK atlas) [25] structures by using the FreeSurfer (Version 6) toolbox in MATLAB R2019b software (MathWorks, Natick, MA, USA). Table 2 summarizes the details of ROIs and abbreviations of the rearranged DK atlas.

### 2.3. FD Analysis and Brain Structural Network

The detailed algorithm for FD is available in numerous previous FD studies [10,11,12,13,14,15]. In brief, the power law relationship defines the FD of a fractal object as:(1)$$N\left(r\right)\alpha r^{-FD}$$
where N(r) denotes the minimal number of cubes of size r covering the fractal object. If then, take the logarithmic operation (to the base 2) on both sides of Equation (1), and the equation can be rewritten in the form of a line as:(2)$$\log_{2}\left(N\left(r\right)\right)=\mathrm{FD}*\log_{2}\left(\frac{1}{r}\right)+k$$
which means that the value of FD can be estimated from the slope of the line.

The first step of the FD calculation process is to select cubic boxes with side length r pixels (set r = 10 pixels as initial value) and stack them side by side to cover the whole 3D fractal object. The count is set to 1 if the box contains any pixels belonging to the fractal object and 0 otherwise. The total number of non-empty boxes is calculated and the result is set to N. The same counting process is repeated by gradually decreasing the size r of the boxes. The number of boxes in the series of N(r) is obtained from r = 10 to r = 2, and the series of Ns and (1/r) is logarithmically counted (with a base of 2). A scatter plot of log_2_ N(r) versus log_2_(1/r) is plotted and the range of box sizes corresponding to the linear portion of the regression line (red and blue lines in Figure 1) that achieves the highest slope correlation coefficient (*R*^2^) is determined. Finally, the slope of the regression line with the highest slope correlation (red line in Figure 1) will be the FD value of the measured object.

### 2.4. Network Property Analysis of Intra-Modular and Inter-Modular Connectivity

There are two steps to build a brain structural network: One step is to calculate the correlation between pairs of subregions to indicate the strength of the inter-regional connectivity. Thus, the brain structural network is derived from the 68 × 68 correlation matrix of the FD values of the paired regions. Secondly, we use a modular analysis to separate the different brain distinctions into modules based on their inter-regional connections [27]. Modularity (Q) indicates the number of edges where all pairs of nodes within the same module. 

The Q can be expressed as:(3)$$Q=\frac{1}{2m}\underset{i,j}{\sum }\left[A_{i,j}-\frac{k_{i}k_{j}}{2m}\right]\delta \left(c_{i},c_{j}\right)$$
where *A* is the connection matrix of the network, and each element of *A* is the correlation coefficient between regions; $k_{i}=\underset{j}{\sum }A_{ij}$ is defined as the sum of the correlation coefficient between node *i* and its connected regions, and is also called the degree of node *I*; $m=\frac{1}{2}\underset{i,j}{\sum }A_{ij}$ represents the total number of edges; and $c_{i}$ denotes the module of node *i*. The δ -function $\delta \left(i,j\right)$ is 1 when nodes *i* and *j* belong to the same module and 0 otherwise. If the investigated network presents superior partitioning, it will have a greater Q value and is more likely to construct a modular organization [21].

In this study, we have set a proportional value of 0.2 as a threshold to filter the connectivity matrix by preserving 20% proportion of the strongest correlation coefficients. In this process, all other entries below the threshold, negative correlations, and all entries on the main diagonal (self-to-self connections) are set to 0 and the links will not exist. 

#### 2.4.1. Intra-Modular Connectivity Analysis

Intra-module and inter-module connections are two important parameters for assessing the density of modular connectivity [28]. Within each module, intra-modular connectivity indicates how densely each node is connected. The intra-modular connectivity (*Z_i_*) is defined as follows:(4)$$Z_{i}=\frac{K_{i}-\bar{K_{ci}}}{\sigma_{K_{ci}}}$$

In Equation (4), *K_i_* is the number of links of *i*th node to other nodes in module *C*; $\bar{K_{ci}}$ is the average of all *K*_i_, and $\sigma_{K_{ci}}$ is the standard deviation of all *K_i_*. Thus, the intra-modular connectivity (*Z_i_*) can indicate the connecting degree of node *i* within the measured module. Then, the mean of all *Z_i_* within each module is the intra-module connectivity. To investigate the functional connecting alterations in the lobes of the MCI group, we took all nodes in each lobe as the within module nodes and calculated the intra-modular connectivity as the intra-lobular connectivity.

#### 2.4.2. Inter-Modular Connectivity Analysis

The inter-modular connectivity shows the strength of the linkage of a given node with other modules. The inter-modular linkage is defined as the participation coefficient, which can be expressed as:(5)$$P_{i}=1-\sum_{c-1}^{c}{\left(\frac{K_{ci}}{K_{i}}\right)}^{2}$$

In Equation (5), the *K_ci_* is the number of links of node *i* in module c, and *K_i_* is the total degree of node *i*. The participation coefficient of a node is, therefore, close to 1 if its links are uniformly distributed among all the modules and 0 if all its links are within its own module. Similarly, the mean of all *P_i_* within each module is the inter-modular connectivity. We took all nodes in each lobe as the within module nodes and calculated the inter-modular connectivity as the inter-lobular connectivity.

### 2.5. Statistical Analysis

In this study, a two-tailed *t*-test and multiple false discovery rate (FDR) correction were used to compare measurements [29], including FD values and brain structural network parameters, between the control and MCI patient groups. Then, the effect size process was applied to measure the strength of the relationships among compared variables of the groups to indicate the practical differences [30]. Note that each group with 30 subjects had 68 FD values for each region. We computed the FD value on the basis of a correlation between any two regions. As a result, a 68 × 68 correlation map was obtained for each group to build a structural network, resulting in one set of topological properties for each structural network. Accordingly, we could not directly perform any statistical comparison of the corresponding topological properties between these two structural networks.

In this study, a permutation test was conducted to statistically compare the differences in network properties between the two groups [31]. To test the null hypothesis, we randomly selected 10 MCI patients and 10 control participants from each study group (20 subjects) and reassigned these 20 subjects as the randomized MCI group and randomized control group, separately. This randomized simulation and recalculation of the network properties were repeated 1000 times to compute the correlation matrix for each randomized group. The 95th percentile points of each distribution of the 1000 simulations were used as critical values in a two-sample, one-tailed *t*-test to reject the null hypothesis, with a type I error probability of 0.05. Then, the network properties Q, P, and Z were calculated for each reassigned correlation matrix of the two groups. Following the permutation process, 1000 sets of network parameters were used in a two-sample, one-tailed *t*-test with FDR correction to assess significant differences between the study groups.

## 3. Results

### 3.1. Patients with MCI Exhibit Significant Lateralized FD Changes Mainly in Temporal Lobe Regions

Table 3 summarizes the FD values of each lobe in the control and MCI groups. The MCI group revealed significantly lower FD values in their bilateral temporal lobes (left temporal, *p* < 0.001 and right temporal, *p* = 0.0025) and right parietal lobe (*p* = 0.0015).

Table 4 summarizes the subregions with significantly decreased FD values in the MCI group. The MCI group showed 27 subregions with significantly decreased FD values, and mainly in the right hemisphere (L/R, 10/17). Among these 27 significantly decreased FD subregions, 11 subregions were in the temporal lobe (L/R, 6/5), 7 subregions in the frontal lobe (L/R, 2/5), 7 subregions in the parietal lobe(L/R, 2/5) and 2 subregions in the right occipital lobe.

### 3.2. Patients with MCI Exhibit Lower Correlation Rates within and between Lobes

Figure 2a,b illustrate the correlation plots between different brain lobes in the control and MCI groups for the subregions. The color bars indicate the correlation rates. First, the control group shows a higher correlation rate range from 0 to 0.8, while the MCI group shows a lower correlation rate from 0 to 0.65. For the control group, the mean correlation rate is 0.3276 for the frontal lobe, 0.2754 for the temporal lobe, 0.3719 for the parietal lobe, and 0.2888 for the occipital lobe. The MCI group shows lower correlation rates for each lobe. They show correlation rates of 0.1679 for the frontal lobe, 0.1861 for the temporal lobe, 0.1451 for the parietal lobe, and 0.1540 for the occipital lobe.

### 3.3. Patients with MCI Reveal Smaller Modular Size and Less Node Integration in Their Brain Structural Network 

Figure 3a,b illustrate the node distribution of modules in the FD-based brain structural network for the control and MCI groups, respectively. The subfigures were plotted using the BrainNet Viewer software [32]. According to the network analysis, the 68 subregions of the cortex in the control group are clustered into five middle segments, whereas the 68 subregions of the cortex in MCI patients are clustered into six modules.

Table 5 summarizes the detailed subregions within each module of the modular brain network for the normal control and MCI groups. In each module, the top row indicates nodes in the left hemisphere and the bottom row indicates nodes in the right hemisphere. In this study, we defined the largest module as the first module, the second largest module as the second module, and so on. The control group has three larger modules (Module 1, 19 nodes; Module 2, 17 nodes; Module 3, 17 nodes), and two smaller modules (Module 4, 8 nodes and Module 5, 7 nodes). The MCI group shows smaller module sizes in their brain network than the control group. There are only one larger module (Module 1, 18 nodes), three medium-sized modules (Module 2, 13 nodes; Module 3, 11 nodes; Module 4, 10 nodes), and two smaller modules (Module 5, 9 nodes and Module 6, 7 nodes) in the MCI group’s brain network. After 1000 permutations were calculated, the network modularity (Q) values were significantly lower in the MCI group than those in the control group (normal, 0.2548 ± 0.0057 and MCI, 0.2451 ± 0.0066, *p* < 0.05, Effect Size = 0.62). This result implies a relatively low density and efficiency of the structural brain network in the MCI group.

For the control group, Modules 1, 2, and 3 all show the integration of nodes from the four functional lobes (frontal, temporal, parietal, and occipital). In Module 1, there are three pairs of bilateral nodes and, in Module 2, there are four pairs of bilateral nodes (bolded in Table 5a). Unlike the control group, the MCI group shows sparse clustering and less functional node integration in the brain structural network modular groupings. In the MCI group, Modules 1, 2, and 3, nodes in the frontal, parietal, and temporal lobes were integrated, but none of the nodes were integrated in the occipital lobe. The MCI group also shows fewer bilateral node pairs in its brain structural network modules. In Module 1, there are two bilateral node pairs linked. In Modules 2 and 3 of the MCI group, there was only one pair of bilateral node links (bolded in Table 5b).

### 3.4. Patients with MCI Reveal Significant Alteration of Intra-Lobular and Inter-Lobular Connectivity in Their Brain Structural Network

Figure 4a–h illustrate the detailed connections within each brain lobe in the control and MCI groups, and Table 6 summarizes the number of connections within the left hemisphere, left hemisphere, and between hemispheres. First, the link distribution in each lobe of the MCI group shows a similar pattern to that of the control group. In each brain lobe, the MCI group shows fewer links and thinner link widths. The MCI group shows the most loss of lateral links in the left temporal lobe (control group, 9 links and MCI group, 4 links) and also the most loss of bilateral links in the temporal lobe (control group, 24 links and MCI group, 14 links). As compared with Figure 4c and g, only two red lines in Figure 4g are wider than those in Figure 4c, which may imply that the MCI group has the most severe decrease in bilateral link strength in the parietal lobe.

Table 7 summarizes the intra-lobular connectivity and connectivity ratios (MCI/control) of cerebral lobes for the control and MCI groups. The MCI group shows significantly decreased intra-lobular connectivity in all cerebral lobes (*p* < 0.05, FDR corrected). Among the four lobes of the MCI group, the parietal lobe reveals the most ratio decreased alteration of intra-lobular connections (81.3%). In the temporal lobes, the MCI group also exhibits the most significant loss in bilateral connections and decreased intra-lobular connectivity with a ratio of 83.5%. 

Table 8 lists the inter-lobular connection rates between the control and MCI groups for each lobe. Figure 5a,b illustrate the inter-lobular nodal connections between the control and MCI groups in the left and right hemispheres. Figure 5c,d illustrate the inter-lobular connections between the bilateral hemispheres of the control and MCI groups. The MCI group exhibits reduced inter-lobular connections in all brain lobes (*p* < 0.05), with the highest rate of reduction in the occipital lobe (*p* < 0.01). In Figure 5b, the MCI group shows that in the right hemisphere, most of the connections are lost from the occipital lobe to the frontal (red line) and parietal (purple line) lobes. This symptom is also shown in Table 7, where the MCI patients show the most reduced inter-lobular connections in the occipital lobe with a rate of 67.7%. The MCI group also shows that there are many nodal connections lost between the right parietal and right temporal lobes. As compared with Figure 5c,d, the connectivity links obtained in the MCI group are similar to those shown in the control group. The reduction of inter-lobular connections was more severe in the lateral hemisphere (right side) and less between the bilateral hemispheres.

## 4. Discussion

In this study, we applied DK atlas to divide the cerebral cortex into 68 subregions and used the FD to measure morphological changes in these divided cortical regions in MCI patients and normal control participants. Based on the correlation maps of the segmented regions, we built an FD-based brain structural network to assess structural network parameters between MCI patients and control participants, including modular, intra-, and inter-lobular connections. The MCI group showed more morphological changes in the right hemisphere (temporal and parietal lobes) than in the left hemisphere (temporal lobes). In addition, they showed a lateral FD values reduction effect, with 17 subregions of FD reduction in the right hemisphere and only 10 subregions of FD values reduction in the left hemisphere.

As a whole, the subregions of the control group were grouped into five modules, whereas in the MCI group, the subregions were grouped into six modules. Lower modularity values and smaller component size modules were detected in the brain network of the MCI group. The MCI group showed lower intra-lobular connectivity in all brain lobes and exhibited the most connections between the bilateral temporal lobes. Normal control participants and the MCI group showed a more similar pattern of inter-lobular connectivity in bilateral connections than in lateral hemisphere connections. The MCI group showed effects of separation, sparser connections and loss of lateral inter-lobular connections, mainly in the right hemisphere. The MCI group showed maintenance of bilateral nodal connections to prevent functional loss of intra- and inter-lobular connections.

### 4.1. FD Analyis Reveals Better Ability for Detecting of Cerebral Changes in MCI Patients

The FD approach is a consistent and the most frequently chosen feature that has been proposed to calculate the intrinsic structural complexity of the cerebral cortex to predict cognitive decline in disease and can complement standard imaging [33]. Traditional methods such as cortical thickness or volume show that MCI patients may exhibit atrophy of their cerebral cortex mainly in the medial temporal, hippocampus, entorhinal, and some sporadic reports in the para hippocampus, amygdala, fusiform gyrus, lateral temporal, parietal, frontal, and occipital lobes [6,7,8]. However, in neurodegenerative diseases, the complexity of assessing cortical shape may better reflect symptoms of atrophy than using traditional volumetric measures [34]. In this study, we prospectively applied FD to measure cortical DK subregions in MCI patients, and the regions of atrophy that we identified included those measured by conventional methods in previous studies [6,7,8], as well as additional subregions in the medial orbital frontal, paracentral, inferior parietal, and superior parietal lobes. Our results showed that the medial temporal, para hippocampal, paracentral, entorhinal, fusiform, postcentral, and superior parietal were the subregions with more decreased FD values in MCI patients.

Using the same FD analysis, Nicolas Nicastro et al. reported that the orbitofrontal cortex and paracentral gyrus were particularly vulnerable in terms of memory and language impairment, and that the FD represented a sensitive imaging marker for prevention and diagnostic strategies [34]. In subjects with MCI, precise measurement of medial temporal lobe atrophy (MTA) may improve predictive accuracy and reduce false-negative classification of dementia [35]. Furthermore, it has been highlighted that visual assessment of MTA on a brain MRI using a standardized rating scale was a strong and independent predictor of conversion to dementia in relatively young MCI patients [36]. With increasing duration of MCI, measuring hippocampal atrophy in older MCI patients has been reported to predict subsequent conversion to AD [37]. Structural abnormalities in the orbitofrontal cortex (OFC) may reflect a potential neurodevelopmental risk marker for MCI [38]. Taken together, our results support these previous findings in MCI and may provide a new approach for identifying MCI.

### 4.2. Patients with MCI Show Shrinkage of Modular Size and Less Functional Lobe Integration in Their Brain Structural Network 

Structural networks are believed to shape and provide constraints for the dynamics of functional connectivity and, to some extent, it has been widely acknowledged that functional networks can be predicted from the underlying structural connectome [39]. A high goodness-of fit level for the structure-function mapping of brain networks has been reported [40,41], as well as a pattern dependence between the connection matrices of the resting-state functional and structural networks [41]. A robust modeular analysis has also reflected a reliable combination of structural and functional networks that were optimally correlated, with the structural network predicting the functional network, but the two networks were not necessarily overlapping [40].

Functional network studies have reported that cerebral subregions that exhibited different combinations of control signals in many tasks could be grouped into three distinct networks, namely the fronto-parietal network (FPN), cingulo-opercular network (CON), and default mode network (DMN) [42,43,44]. The FPN includes the prefrontal, middle cingulate gyrus, inferior parietal, and precuneus; the CON includes the prefrontal, insula, anterior cingulate, and superior frontal lobes; and the DMN includes the inferior temporal, para hippocampal, lateral parietal, and posterior cingulate gyrus [42].

In the present study, we investigated and compared the brain structural network patterns in normal control group and an MCI group. In the control group, we found network Module 1 included many of the nodes of the FPN (frontal, parietal, and precuneus). Network Module 2 included many of the nodes of the DMN (para hippocampal (left and right) and posterior cingulate (left and right)). Network Module 3 included many nodes of the CON (anterior cingulate, superior frontal, and insula).

In the MCI group, we found that the MCI group showed a lower correlation ratio in the correlation maps, which resulted in a separation of the modular groupings. The control group showed five modules in the brain structural network, whereas the MCI group had six modules in their brain network. The MCI group showed a reduction in module size, fewer integrated components, and significantly reduced overall modularity of their brain structural network. Thus, the MCI group showed significantly lower modularity values than those shown by the normal control group. In the major modules of both groups (Modules 1, 2, and 3), the normal control group had distribution points scattered over a larger area and showed denser connectivity between each node than the MCI group. In Module 1, the MCI group exhibited the effect of occipital lobe separation, whereas in Module 3 of the MCI group, the included nodes showed many of the same overlapping nodes as in Module 1 of the normal control group.

Previous MCI diffusion tensor imaging (DTI) studies have shown that possible MCI in the posterior brain suffered from white matter abnormalities and showed significantly reduced anisotropy (FA) in the cuneus, fusiform, peripheral, and occipital lobes [45]. Abnormalities in these regions may lead to functional segregation and may reduce the strength of connectivity of brain networks. It has been reported that the functional network of normal subjects, including the insula, known as the largest homologous module, lost symmetrical functional connectivity properties and the corresponding gray matter concentration (GMC) was significantly reduced in an AD group [46]. Similarly, our results reported significant atrophy of the fusiform and pericalcarine in the MCI group, while insults were included in Module 1 in the normal control group. We further reported that in Module 1 of the MCI group, the bilateral occipital lobes showed a loss of contact with other brain lobes, as shown in previous studies. Our results and other studies have reported significantly reduced network efficiency and properties in MCI groups [47]. Based on our results and other network analyses of MCI, and the high conversion rate from MCI to Alzheimer’s disease (AD), it is foreseeable that an analysis of brain networks could be a suitable tool for detecting MCI and AD [48].

### 4.3. Intra-Lobular and Inter-Lobular Connectivity Decrease in the Brain Structural Network of Patients with MCI

To the best of our knowledge, there are few studies on intra- and inter-lobular connectivity in MCI patients. In this study, we found that MCI patients showed laterally significant decreases in intra-lobular connectivity in all brain lobes, especially in the temporal lobe. The parietal lobe showed the greatest rate of decline, while the temporal lobe showed the most pronounced loss of connections. In the inter-lobular analysis, the MCI group showed a more similar pattern of bilateral hemisphere links as compared with that revealed by the control group. For the MCI group, the reduction in lateral hemisphere links between lobes was more severe than the reduction in bilateral hemispheres links.

Brain connectivity patterns in MCI and AD have shown a decreasing trend in inter- and intra-hemispheric connectivity as reported by the metabolic network. Alterations in both frontal-occipital and parietal-occipital connectivity patterns in the metabolic network have been reported to be key features to distinguish AD disease groups [49]. Our study found the same results with loss of frontal-occipital connecting links and parietal-occipital connecting links, as shown in Figure 5a,b. Between cerebral lobes, MCI patients may maintain bilateral hemisphere nodal connections to reduce the loss of connectivity in the lateral hemisphere.

## 5. Conclusions

First, the FD and DK atlas method proposed in this study allows an accurate assessment of altered brain complexity in patients with MCI. Patients with MCI had significantly lower FD values in the bilateral temporal lobes and the right parietal lobe, including the medial temporal, fusiform, and para hippocampal, as well as medial orbitofrontal lobes. Second, lower modularity values, smaller modules, and less subregion integration were found in the FD-based brain networks of MCI patients. They had lower intra-lobular connectivity in all brain lobes, especially in the temporal lobe. The MCI group showed a separation of inter-lobular connectivity and sparser lateral connections more so in the right hemisphere. Between cerebral lobes, the MCI patients may maintain bilateral hemispheres nodal connections to reduce the loss of connectivity in the lateral hemispheres. In conclusion, our method and results presented in this study could be a suitable tool for early detection of MCI.

## Figures and Tables

**Figure 1 brainsci-13-00093-f001:**
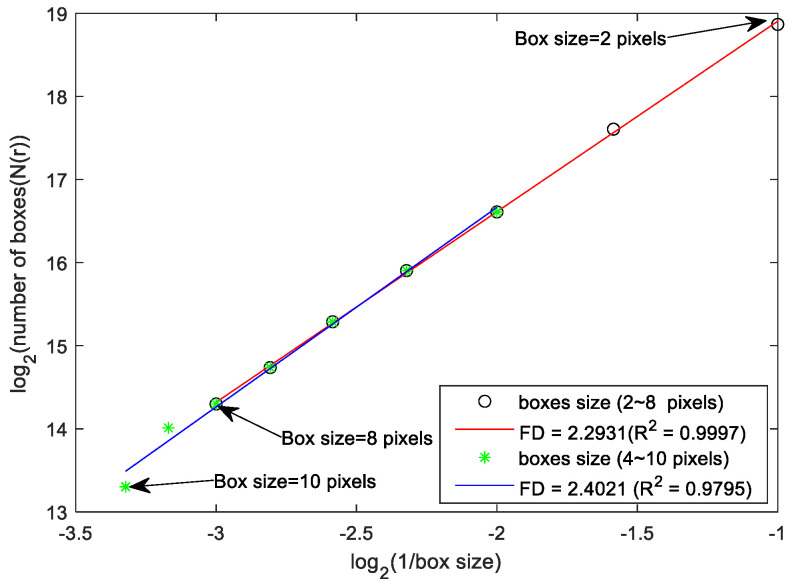
FD measure estimation. Log-log scatter plot, where the x and y axis denote the inverse of the box size and the number of boxes in the logarithmic scale, respectively. The FD value is the slope of the fitting line.

**Figure 2 brainsci-13-00093-f002:**
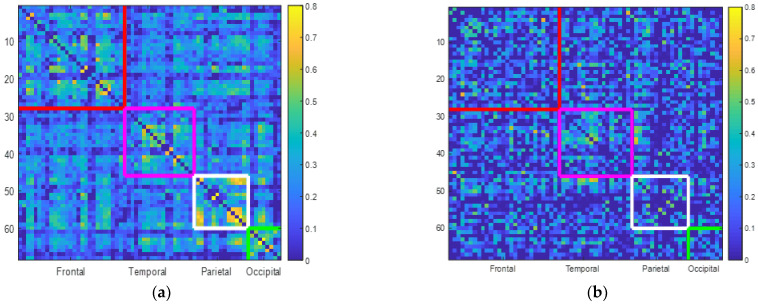
Subregion correlation maps between different brain lobes: (**a**) in the control group and (**b**) in the MCI group. There are 68 rows and 68 columns in each plot, and the dots from the first row and first column from the top left indicate the correlation rate of the first subregion of the ROI (left anterior cingulate) with the other 67 subregions, the second row and second column from the top left indicate the correlation rate of the second subregion of the ROI (right anterior cingulate) with the other 67 subregions, and so on. In each figure, subregions of the frontal lobe are labeled within the red line, the temporal lobe is labeled within the purple line, the parietal lobe is labeled within the white line, and the occipital lobe is labeled within the green line. The color bars indicate the density of correlation, and the color bars of the control group are scaled higher than those of the MCI group. For each brain lobe, normal control participants showed higher correlation densities within the lobes and with other lobes than the MCI group.

**Figure 3 brainsci-13-00093-f003:**
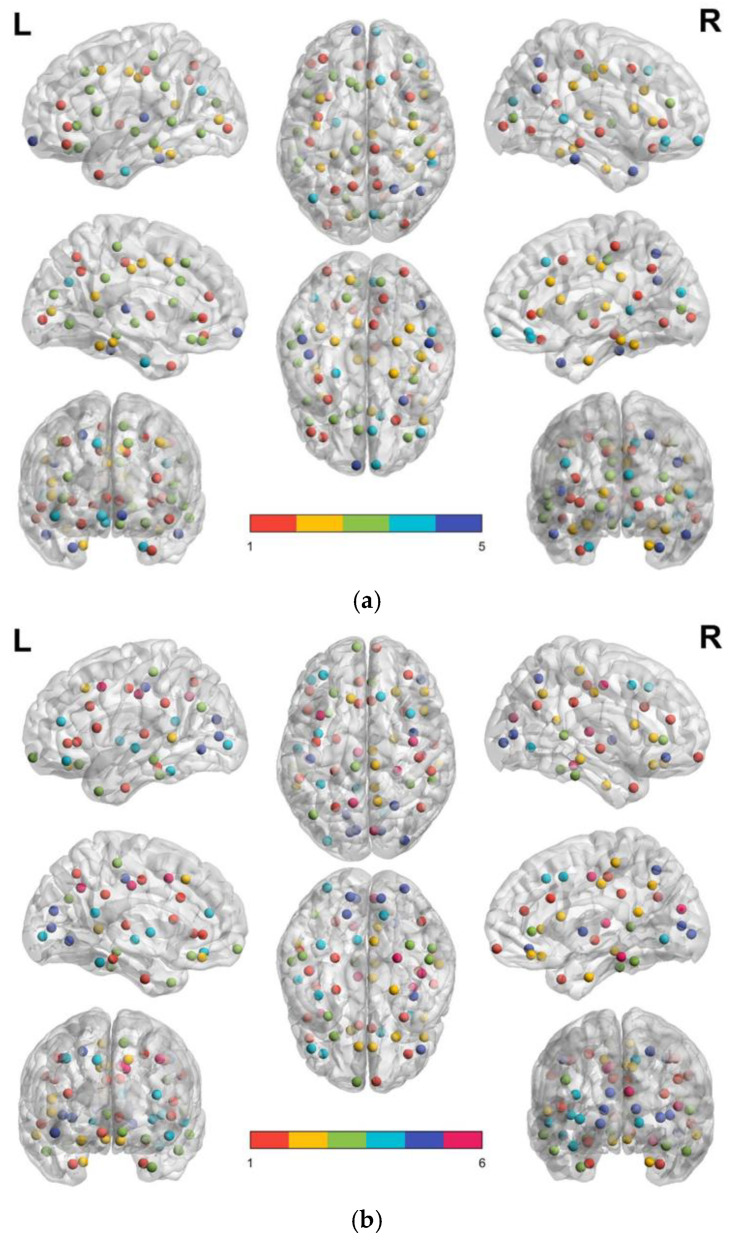
The node distribution of modules in the FD-based brain structural network for the control and MCI groups.(**a**) The distribution of cerebral subregions in each modules of control participants group; (**b**) the distribution of cerebral subregions in each modules of brain structural network of patients with MCI. In each subfigure, red dots demonstrate the nodes of Module 1, yellow dots for Module 2, green dots for Module 3, turquoise dots for Module 4, and royal blue dots for module 5.

**Figure 4 brainsci-13-00093-f004:**
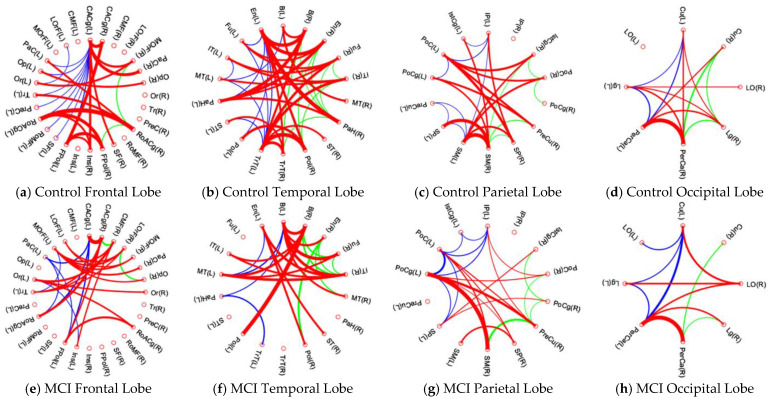
Intra-modular connectivity of each lobe in the control group and MCI group. Control group: (**a**) frontal lobe; (**b**) temporal lobe; (**c**) parietal lobe; (**d**) occipital lobe. MCI group: (**e**) frontal lobe; (**f**) temporal lobe; (**g**) parietal lobe; (**h**) occipital lobe. In each figure, the blue lines depict the lateral links within the left cerebral hemisphere, while the green lines depict the lateral links within the right cerebral hemisphere. The red lines depict the bilateral links between left and right hemispheres. In each figure, the wider line implies a stronger connecting strength, and the thinner line implies a lower connecting strength.

**Figure 5 brainsci-13-00093-f005:**
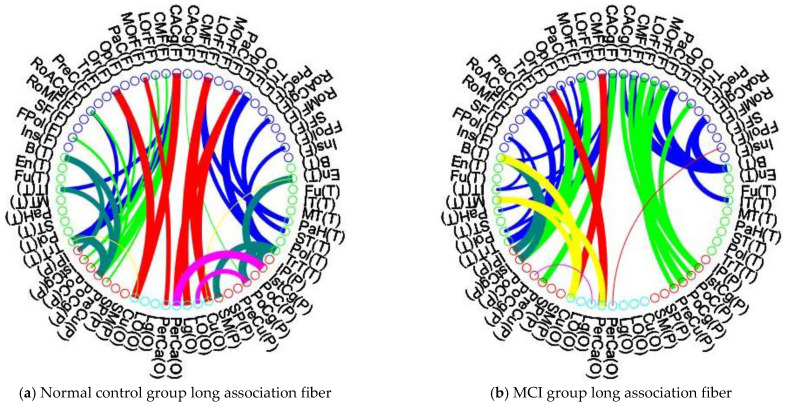
Long lateral hemisphere connections and bilateral hemisphere connections between lobes of the control and MCI groups: (**a**) Control group long lateral hemisphere connections; (**b**) MCI group long lateral hemisphere connections; (**c**) control group bilateral hemispheres connections; (**d**) MCI group bilateral hemispheres connections. In each figure, the left half depicts the left hemisphere, while the right half depicts the right hemisphere. In each hemisphere, there are four lobes, labeled frontal, temporal, parietal, and occipital. Each ROI of the frontal lobe is labeled with a blue circle and abbreviation, the temporal lobe is indicated with a green circle, the parietal lobe with a red circle, and the occipital lobe is illustrated with a green-blue circle. As revealed in the intra-lobular connectivity analysis, the width of the connecting lines in each figure indicates the strength of the connectivity coefficient between nodes, with wider lines implying higher connectivity between nodes. In Figure 5, we use the color of the lines to indicate the different connecting links between the brain lobes. Blue lines indicate links from frontal to temporal lobes, frontal to parietal connections are represented by yellow-green lines, and links from frontal to orbital lobes are represented by red lines. Yellow lines indicate links from the temporal lobe to the occipital lobe, connections between the temporal and parietal lobes are represented by mossy green lines, and purple lines indicate connections between the occipital and parietal lobes.

**Table 1 brainsci-13-00093-t001:** Sample demographics and characteristics of control participants and the MCI group.

	MCI (*n* = 30)	CP (*n* = 30)	*p* Values
Age, yearsAge, years	70 ± 5.2	69 ± 3.7	0.57
Sex, female/male	16/14	15/15	0.8003
Dominant hand, right/ left	30/0	30/0	
Education, years	11.3 ± 3.6	11.4 ± 3.4	0.558
MMSE	24.76 ± 3.22	28.86 ± 0.74	<0.001
CDR global	0.5	0	
CDR Memory	0.5	0	

MCI, mild cognitive impairment; CP, control participants; MMSE, mini-mental state examination; CDR, clinical dementia rating scale; *p*-value was obtained using a two-sample two-tail *t*-test.

**Table 2 brainsci-13-00093-t002:** Regions of interest (ROIs) and abbreviations of the rearranged Desikan–Killiany atlas. ROIs of the frontal lobe (1-28, odd numbers denote the ROIs in the left hemisphere and even numbers denote the ROIs in the right hemisphere), temporal lobe (29–46), parietal lobe (47–60), and occipital lobe (61–68) are shown.

Frontal	ROI	Abbreviation	Temporal	ROI	Abbreviation
1,2	Caudal anterior cingulate	CACg	37,38	Medial temporal	MT
3,4	Caudal middle frontal	CMF	39,40	Para hippocampal	PaH
5,6	Lateral orbital frontal	LOrF	41,42	Superior temporal	ST
7,8	Medial orbital frontal	MOrF	43,44	Temporal pole	TPol
9,10	Paracentral	PaC	45,46	Transverse temporal	TrT
11,12	Parsopercularis	Op	Parietal		
13,14	Parsorbitalis	Or	47,48	Inferior parietal	IP
15,16	Parstriangularis	Tr	49,50	Isthmus cingulate	IstCg
17,18	Precentral	PreC	51,52	Postcentral	PoC
19,20	Rostral anterior cingulate	RoACg	53,54	Posterior cingulate	PoCg
21,22	Rostral middle frontal	RoMF	55,56	Precuneus	PreCu
23,24	Superior frontal	SF	57,58	Superior parietal	SP
25,26	Frontal pole	FPol	59,60	Supra marginal	SM
27,28	Insula	Ins	Occipital		
Temporal			61,62	Cuneus	Cu
29,30	Bankssts	B	63,64	Lateral occipital	LO
31,32	Entorhinal	En	65,66	Lingual	Lg
33,34	Fusiform	Fu	67,68	Pericaicarine	PerCa
35,36	Inferior temporal	IT			

**Table 3 brainsci-13-00093-t003:** FD measurement results of cerebral lobes in control and MCI groups.

Lobe	Control	MCI	*p*-Value	Effect Size
Frontal (L)	2.2324 ± 0.0161	2.2319 ± 0.0224	0.9149	0.0128
Frontal (R)	2.2393 ± 0.0144	2.2331 ± 0.0213	0.1858	0.1681
Temporal (L)	2.2108 ± 0.0108	2.1908 ± 0.0200	<0.001 *	0.5283
Temporal (R)	2.1985 ± 0.0146	2.1849 ± 0.0189	0.0025 *	0.3735
Parietal (L)	2.2902 ± 0.0183	2.2844 ± 0.0158	0.1922	0.1672
Parietal (R)	2.2990 ± 0.0184	2.2822 ± 0.0209	0.0015 *	0.3924
Occipital (L)	2.2146 ± 0.0301	2.2084 ± 0.0299	0.4262	0.1028
Occipital (R)	2.2247 ± 0.0334	2.2159 ± 0.0240	0.2469	0.1498

**Table 4 brainsci-13-00093-t004:** Cortical subregions of significantly decreased FD values in the MCI group.

	Left Hemisphere	Controls	MCI	*p*-Value	Effect Size
Frontal (L)	Paracentral ±	2.1765 ± 0.0523	2.1546 ± 0.0443	0.0268	0.22
	Rostral middle frontal	2.4095 ± 0.0178	2.4009 ± 0.0247	0.0353	0.20
Frontal (R)	Caudal anterior cingulate	2.1322 ± 0.0554	2.1037 ± 0.0497	0.0196	0.26
	Caudal middle frontal	2.2911 ± 0.0448	2.2766 ± 0.0335	0.0363	0.18
	Medial orbital frontal	2.2473 ± 0.0359	2.2235 ± 0.0776	0.0342	0.19
	Paracentral	2.2118 ± 0.0361	2.1724 ± 0.0378	0.0003	0.47
	Superior frontal	2.4082 ± 0.0169	2.3988 ± 0.0166	0.0189	0.27
Temporal (L)	Medial Temporal	2.3148 ± 0.0274	2.2970 ± 0.0334	0.0245	0.28
	Fusiform	2.2993 ± 0.0271	2.2853 ± 0.0248	0.0183	0.26
	Inferior temporal	2.3358 ± 0.020	2.3223 ± 0.0294	0.0182	0.26
	Transverse Temporal	2.0486 ± 0.0498	2.0229 ± 0.0595	0.0248	0.23
	Entorhinal	2.1278 ± 0.0415	2.0766 ± 0.0518	0.0004	0.47
	Temporal pole	2.1937 ± 0.0352	2.1734 ± 0.0445	0.0213	0.25
Temporal (R)	Bankssts	2.1815 ± 0.0385	2.1622 ± 0.0562	0.0352	0.20
	Fusiform	2.2994 ± 0.0268	2.2816 ± 0.0341	0.0195	0.28
	Para hippocampal	2.0464 ± 0.0590	2.0165 ± 0.050	0.0200	0.26
	Transverse Temporal	1.9913 ± 0.0621	1.9542 ± 0.0654	0.0216	0.28
	Superior temporal	2.3435 ± 0.0351	2.333 ± 0.0281	0.0438	0.16
Parietal (L)	Postcentral	2.2919 ± 0.02879	2.2807 ± 0.0267	0.0368	0.20
	Supra marginal	2.3867 ± 0.0271	2.3737 ± 0.0263	0.0242	0.28
Parietal (R)	Inferior parietal	2.4292 ± 0.020	2.4181 ± 0.0171	0.0275	0.27
	Superior parietal	2.3581 ± 0.0221	2.3424 ± 0.0193	0.0076	0.35
	Postcentral	2.2829 ± 0.02169	2.2624 ± 0.0279	0.0047	0.38
	Poster cingulate	2.2097 ± 0.0602	2.1864 ± 0.0652	0.0362	0.18
	Supra marginal	2.3886 ± 0.02437	2.3574 ± 0.0329	0.0342	0.47
Occipital (R)	Lateral occipital	2.3713 ± 0.0256	2.3619 ± 0.0261	0.0357	0.18
	lingual	2.2721 ± 0.0257	2.2528 ± 0.0373	0.0288	0.29

**Table 5 brainsci-13-00093-t005:** The detailed subregions within each module of the modular brain network for the normal control and MCI groups. (a) Modules of FD-based brain structural network in the control group (b) Modules of FD-based brain structural network in the MCI group. Bold in the table indicates the pair of bilateral node links in the same module.

(a)	Modules of FD-based brain structural network in the control group
Module 1 (19)	Frontal: parsorbitalis, **roatral anterior cingulate**, rostral middle frontal, insula, Temporal: temporal pole, Parietal:postcentral, **precuneus**, superior parietal, Occipital: **lateral occipital**
	Frontal: **rostral anterior cingulate**, caudal middle frontal, lateral orbital frontal, paracentral Temporal: middle temporal, superior temporal Parietal: isthmus cingulate, **precuneus** Occipital: **lateral occipital**, lingual
Module 2 (17)	Frontal: caudal middle frontal, **precentral**, Temporal: **fusiform**, **parahippocampal**, Parietal: isthmus cingulate, **posterior cingulate**, Occipital: pericalcarine
	Frontal: **precentral**, caudal anterior cingulate, parsopercularis, parstriangularis, Temporal: entorhinal, **fusiform**, **parahippocampal**, transverse temporal Parietal: **posteror cingulate**, supra marginal
Module 3 (17)	Frontal: caudal anterior cingulate, lateral orbital frontal, medial orbital frontal, paracentral, parsopercularis, parstriangularis, superior frontal, Temporal: bankssts, middle temporal, superior temporal, Parietal: supra marginal, Occipital: cuneus, lingual
	Frontal: rostral middle frontal, insula, Parietal: postcentral Occipital: pericalcarine
Module 4 (8)	Temporal: entorhinal, Parietal: inferior parietal
	Frontal: medial orbito frontal, parsorbitalis, superior frontal, frontal pole Temporal: bankssts, Occipital: cuneus
Module 5 (7)	Frontal: frontal pole, Temporal: inferior temporal, transverse temporal
	Temporal: inferior temporal, temporal pole, Parietal: superior parietal, inferior parietal
(b)	Modules of FD-based brain structural network in the MCI group
Module 1 (18)	Frontal: **caudal anterior cingulate**, parsopercularis, parstriangularis, rostral anterior frontal, Temporal: entorhinal, para hippocampal, transverse temporal, Parietal: precentral, superior parietal, **supra marginal**
	Frontal: **caudal anterior cingulate**, rostral middle frontal, frontal pole, Temporal: superior temporal, temporal pole, Parietal: inferior parietal, postcentral, **supra marginal**
Module 2 (13)	Frontal: **medial orbital frontal**, superior frontal, Temporal: bankssts, entorhinal
	Frontal: lateral orbital frontal, **medial orbital frontal**, paracentral, parsopercularis, parstriangularis, Temporal: middle temporal, Parietal: isthmus cingulate, posterior cingulate, precuneus
Module 3 (11)	Frontal: lateral orbital frontal, paracentral, frontal pole, Temporal: **inferior temporal**, middle temporal, temporal pole, Parietal: inferior parietal
	Frontal: rostral anterior frontal, Temporal: bankssts, fusiform, **inferior temporal**
Module 4 (10)	Frontal: parsorbitalis, rostral middle frontal, insula Temporal: fusiform, superior temporal Parietal: isthmus cingulate, Occipital: lateral occipital
	Frontal: caudal middle frontal, superior frontal, Occipital: lingual
Module 5 (9)	Parietal: postcentral, Occipital: cuneus, lingual, **pericalcarine**
	Frontal: parsorbitalis, insula, Parietal: superior parietal, Occipital: lateral occipital, **pericalcarine**
Module 6 (7)	Frontal: caudal middle frontal, Parietal: posterior cingulate, precuneus
	Frontal: precentral, Temporal: para hippocampal, transverse temporal, Occipital: cuneus

**Table 6 brainsci-13-00093-t006:** The number of links between nodes in the lateral lobe and between the bilateral lobes of the control and MCI groups.

Lobe	Frontal	Temporal	Parietal	Occipital
Hemisphere	L/R/B	L/R/B	L/R/B	L/R/B
Controls	9/2/15	9/8/24	7/5/13	3/3/8
MCIs	6/2/18	4/6/14	7/4/12	4/2/5

L/R/B, links in left hemisphere/ links in right hemisphere/ links between bilateral hemispheres.

**Table 7 brainsci-13-00093-t007:** The intra-lobular connectivity and connectivity ratios (MCI/controls) of four cerebral lobes for the control and MCI groups. ***** *p* < 0.05.

Group	Frontal	Temporal	Parietal	Occipital
Controls	0.3686	0.4239	0.3378	0.4642
MCIs	0.3157 *	0.3542 *	0.2747 *	0.4125 *
Ratio (MCI/Control)	85.6%	83.5%	81.3%	88.8%

**Table 8 brainsci-13-00093-t008:** The inter-lobular connectivity between the lobes of the control and MCI groups. ***** *p* < 0.05 and ****** *p* < 0.01.

Group	Frontal	Temporal	Parietal	Occipital
Controls	0.6345	0.633	0.6265	0.6341
MCIs	0.5440 *	0.5558 *	0.4772 *	0.4293 **
Ratio (EM/Control)	85.7%	87.8%	76.2%	67.7%

## Data Availability

Patient consent was waived due to the privacy concern raised by our IRB.

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
