# Peer review of "Decreased Brain Structural Network Connectivity in Patients with Mild Cognitive Impairment: A Novel Fractal Dimension Analysis"

_brainsci, 2023, doi:10.3390/brainsci13010093_

Round 1

Reviewer 1 Report

This paper try to show the FD technique for the study of the MCI and its comparison with a control group. The conception is interesting and the FD results produce a working environment that allows the detection of neurofunctionally interesting activations. The idea of ​​being able to model the transit between MCI and AD continues to be an important topic. In my opinion, there are some limitations that must be taken into account, beyond the recurring sample size.

1. It is not clearly mentioned if the sampling has been matched and what variables have been taken into account for it. There is talk of sociodemographic variables but it is not clear whether they are matched between groups.

2. The use of a 1.5T scanner is an extremely important limitation. Currently the minimum should be 3T

3. The aspect that most hinders an orderly reading of the work is the continuous confusion between the results section and the conclusions. the presentation of the results already supposes a list of conclusions so that the reader anticipates how those results are interpreted. Suvede, then, that the conclusions are reiterative and do not go into detail. The conclusions offered in the results are better, in my opinion, than in the conclusions themselves. These two sections must be rearranged to clearly separate one thing from another.

Author Response

Brainsci-2004189

Reviewer #1

This paper try to show the FD technique for the study of the MCI and its comparison with a control group. The conception is interesting and the FD results produce a working environment that allows the detection of neurofunctionally interesting activations. The idea of ​​being able to model the transit between MCI and AD continues to be an important topic. In my opinion, there are some limitations that must be taken into account, beyond the recurring sample size.

  1. It is not clearly mentioned if the sampling has been matched and what variables have been taken into account for it. There is talk of sociodemographic variables but it is not clear whether they are matched between groups.

Author’s response: We are very grateful to the reviewers for their valuable suggestions. In the revised manuscript, we recruited an additional 10 MCI and 10 healthy controls as the study group. We have also added a table summarizing the demographics and characteristics of the control and MCI groups in this study. The revised manuscript and the added table are shown below.

“Table 1 summarizes the demographics and characteristics of control participants and MCI groups in this study. A total of 30 patients with MCI (male/female: 14/16) and 30 (male/female:15/15) healthy controls participated in this study.

Table 1 Sample demographics and characteristics of control participants and MCI groups

MCI (n=30)

CP (n=30)

p values

Age, years

70+5.2

69+3.7

0.57

Sex, female/male

16/14

15/15

0.8003

Dominant hand, right/ left

30/0

30/0

Education, years

11.3+3.6

11.4+3.4

0.558

MMSE

24.76+3.22

28.86+0.74

< 0.001

CDR global

0.5

0

CDR Memory

0.5

0

MCI, mild cognitive impairment; CP, control participants; MMSE, mini-mental state examination; CDR, clinical dementia rating scale; p value was obtained using a two-sample two-tail t test.

  1. The use of a 1.5T scanner is an extremely important limitation. Currently the minimum should be 3T

Author’s response: we thank reviewer’s suggestion. The thickness of the MRI slice is very important for FD measurements. If the thickness of the slice is greater than the size of the box used to calculate the FD, then it is not possible to obtain a correct FD value. The thickness of the slice must be smaller than the size of the counting box. In this study, the minimum size of the counting box was 2 mm, while the thickness of the MRI slice was 1.5 mm. Therefore, in this study, the FD calculation was appropriate for patients with MCI and had an accurate result. Although we used a 1.5T scanner in this study, this may interfere with the quantification of MR images to some extent. To ensure the quantification of MR images, we examined all image data slice by slice before further image processing and ensured that all MR images were suitable for FD measurements.

  1. The aspect that most hinders an orderly reading of the work is the continuous confusion between the results section and the conclusions. the presentation of the results already supposes a list of conclusions so that the reader anticipates how those results are interpreted. Suvede, then, that the conclusions are reiterative and do not go into detail. The conclusions offered in the results are better, in my opinion, than in the conclusions themselves. These two sections must be rearranged to clearly separate one thing from another.

Author’s response: We thank the reviewers for their valuable suggestions. In the revised manuscript, we have rephrased and shortened the results section. We have added a table to provide more quantitative analysis and also replaced some graph property descriptions in the figure captions. In the discussion, we have reorganized the paragraphs and added a paragraph discussing the relationship between structural and functional brain networks.

The added table in the revised result section is as follows:

The added paragraph in the revised discussion section is as follows:

4.2. MCI show shrinkage modular size and less functional lobes integration in their brain structural network

Structural networks are believed to shape and provide constraints for the dynamics of functional connectivity, to some extent, it has been widely acknowledged that functional networks can be predicted from the underlying structural connectome [38]. A high goodness of fit level for the structure-function mapping of brain networks has been reported [39,40], as well as a pattern dependence between the connection matrices of the resting-state functional and structural networks [40]. Robust model analysis also reflects a reliable combination of structural and functional networks that are optimally correlated, with the structural network predicting the functional network, but the two networks not necessarily overlapping [39].

Functional network studies have reported that cerebral sub-regions that exhibit different combinations of control signals in many tasks can be grouped into three distinct networks, namely the Fronto-Parietal network (FPN), Cingulo-Opercular network (CON) and default model network (DMN) [41-43]. The FPN includes the prefrontal, middle cingulate gyrus, inferior parietal and precuneus, the CON includes the prefrontal, insula, anterior cingulate and superior frontal lobes, and the DMN includes the inferior temporal, para hippocampal, lateral parietal and posterior cingulate gyrus [41].

In the present study, we investigated and compared the brain structural network patterns in normal controls and MCI group. In the control group, we found network model 1 included many of the nodes of the FPN (frontal, parietal and precuneus). Network model 2 included many of the nodes of the DMN (para hippocampal (Left and right) and posterior cingulate (left and right)). Network model 3 included many nodes of the CON (anterior cingulate, superior frontal and insula).

The added references are as follow:

  1. O’Neill, G.C., Tewarie, P., Vidaurre, D., Liuzzi, L., Woolrich, M.W., Brookes, M.J., 2017. Dynamics of large-scale electrophysiological networks: a technical review. Neuroimage 180, part B, 559–576.
  2. Miši´c, B.; Betzel, R.F.; De Reus, M.A.; Van Den Heuvel, M.P.; Berman, M.G.; McIntosh, A.R.; Sporns, O. Network-level structure-function relationships in human neocortex. Cereb. Cortex 2016, 26, 3285–3296.
  3. Meier J, Tewarie P, Hillebrand A, Douw L, van Dijk BW, Stufflebeam SM, Van Mieghem P. A Mapping Between Structural and Functional Brain Networks. Brain Connect. 2016 May;6(4):298-311. doi: 10.1089/brain.2015.0408. Epub 2016 Mar 29. PMID: 26860437; PMCID: PMC4939447.
  4. Fair DA, Cohen AL, Power JD, Dosenbach NUF, Church JA, et al. (2009) Functional Brain Networks Develop from a ‘‘Local to Distributed’’ Organization. PLoS Comput Biol 5(5): e1000381. doi:10.1371/journal.pcbi.1000381

Other improvement of our revised manuscript, please refer to our revised manuscript.

  1. Dosenbach NU, Fair DA, Miezin FM, Cohen AL, Wenger KK, et al. (2007) Distinct brain networks for adaptive and stable task control in humans. Proc Natl Acad Sci U S A 104: 11073–11078.
  2. Dosenbach NU, Visscher KM, Palmer ED, Miezin FM, Wenger KK, et al. (2006) A core system for the implementation of task sets. Neuron 50: 799–812.

Please refer to our revised manuscript to review the improvements to our manuscript.

Reviewer 2 Report

Lau and colleagues assess regional morphological differences between 20 healthy older adult controls and 20 mild cognitive impairment (MCI) participants using a fractal dimensionality (FD) approach. This analysis revealed reductions in FD in the MCI participants, primarily in temporal regions, but also in limited frontal, parietal, and occipital ROIs. They further use FD to define group-level “networks” by examining patterns of correlation in regional FD values in MCI and controls. They note that MCI network is marked by lower correlations within and between anatomical lobes, and also exhibits less modular organization.

This paper is certainly novel and ambitious. However, there are serious flaws and limitations in the justification / motivation, sample, statistical analyses, and interpretation of results. A summary of my major concerns is below, followed by a more detailed list of my complete comments:

First, as far as I can tell, this is the first application of a graph theoretic / network approach to the fractal dimensionality metric. Hence, validation of this network approach is severely lacking. For instance, how do we know that FD-based “networks” are even reliable / reproducible in healthy control samples? If not, then the differences we are seeing between MCI and controls may reflect noise or sampling variability, rather than any meaningful biological difference. Further, to what extent do FD-based networks capture redundant and/or novel signal compared to established MRI-based network analyses, e.g., resting state functional connectivity (FC) or diffusion tensor imaging (DTI)? Finally, how are these FD-based networks organized in healthy younger adults? The authors use the Desikan-Kiliany parcellation and anatomical lobe assignments in an attempt to capture this organization, but this is very likely not optimal. For comparison, FC and DTI-based network analyses have addressed each of these questions and developed specific parcellations that are optimized for the particular demands of each modality (e.g., for summary, see Power et al., 2014). Similar validations of the FD-based network approach are necessary in large samples of healthy controls, which could be done using large, publicly-available datasets (e.g., Human Connectome Project). Until then, questions regarding group differences in MCI are premature.

Second, the sample of 20 MCI and 20 controls is very small, especially given the concerns noted above regarding the lack of prior demonstrations of the validity and reliability of the FD network approach. The authors cite a prior study that used ADNI data to compare AD vs. controls with a similar FD approach. I see no reason why these analyses could not be performed in a similarly large, public dataset. This would provide a much needed validation of the results in an independent and larger sample.

Third, the authors apply inappropriate statistical tests. In particular, the implementation of the permutation test, in combination with a t test, is likely invalid. I provide detailed suggestions below for how to properly apply the permutation test below. Moreover, the authors often provide qualitative descriptions of the results (i.e., describing the colors of their heatmaps) rather than providing easily quantifiable summary measures and statistics. Finally, in the other analyses, the authors do not provide effect sizes, nor do they address the problem of multiple comparisons in their regional analyses.

Fourth, the authors consistently over-interpret the FD data and network-based results, suggesting that these measures reflect “atrophy”, “fiber links”, “commissural fibers”, and “functional integration”. These interpretations appear to be conflations with other MRI-based volumetric and network analyses and are not warranted by these results or any of the prior work cited.

A complete, detailed list of comments is below:

1.     Abstract: what is meant by “cerebral morphological alteration in detail region”? The authors should be more explicit.

2.     Abstract: Line 25 is also unclear. The complexity measure is “superior” to what?

3.     Abstract: Line 30: Please use terms like “control participants” or “cognitively normal sample”, etc., instead of simply “normal.”

4.     Line 46: MCI is typically considered a “prodromal” stage of AD. “Preclinical” often refers to cognitively normal participants with present amyloid.

5.     Line 50: MCT -> MCI?

6.     Participants: Please provide a table summarizing key demographic details in the MCI and control participants (age, sex, education, any available neuropsychological testing / cognitive screening measures).

7.     The sample of 20 MCI and 20 controls is rather small. I see no reason why these analyses couldn’t be performed in a larger, publicly available dataset (e.g., ADNI). In fact, the authors cite a prior study that appears to have applied a similar FD analysis in ADNI (King et al., 2010). In the interest of validating these results, I would suggest replicating the analyses in a larger, independent cohort.

8.     Line 129: Which version of FreeSurfer was implemented?

9.     Line 146: What is the justification for setting the initial value to r = 10?

10.  Line 152 and Figure 1 indicate that only a subset of the box sizes are considered when finding the best fitting regression line (i.e., the blue line and red line have different ranges). The justification and potential implications of this choice are not clear. For instance, why not fit a regression to the full range and calculate its slope? How much heterogeneity is there in the ranges of selected box sizes between different regions? Between different participants?

11.  Line 163: What values are being correlated in this approach? Presumably this is referring to FD values across individual participants, but this is not clearly stated.

12.  Line 163: The definition of modularity (Q) is unclear as stated. Please be more explicit and provide the equation that was used.

13.  Line 170: What threshold was used to define edges and modules in the current analyses is the justification of this threshold? Are the results consistent across a range of threshold values? This is of course a very common concern for network analyses.

14.  Line 190: “A smaller number of participation coefficient indicates a large degree of intra-modal connectivity within node i”. Is this true? Shouldn’t a smaller participation coefficient simply indicate a smaller degree of inter-modal connectivity at node i?

15.  Statistical analyses: The authors are testing for regional differences in FD in 68 ROIs, and thus increase the type 1 error rate. It is necessary to apply appropriate correction for multiple comparisons.

16.  Line 209: For the permutation test, why were only 10 MCI participants and 10 controls reassigned? What happens to the other 10 participants in each group - are they fixed across permutations? If so, this is an inappropriate implementation. If not, I see no reason why the full sample of 20 MCI and 20 controls shouldn’t be randomly reassigned, as is done in a typical permutation / randomization test. Further 100 permutations seems a little small. I would suggest at least 1000. Finally, this paragraph seems to contain several redundant sentences.

17.  Line 215: It is inappropriate to run a t test on bootstrapped permutations. This creates a problem such that non-meaningful differences would be considered significant simply by adding more and more permutations. Instead, the appropriate test is to generate an “empirical p value” by calculating the proportion of permutations in which an effect (i.e., difference between group 1 and group 2) as or more extreme than the observed difference. For demonstration, see, e.g.: https://rpubs.com/valentin/permutation-test-1 or https://towardsdatascience.com/how-to-use-permutation-tests-bacc79f45749

18.  Line 218: The authors interpret reductions in FD values to reflect “atrophy”. What is the justification for this interpretation? How do these measures compare to more established indicators of “atrophy”, namely regional volume and cortical thickness?

19.  Results: It would be very helpful to report the regional FD values visually plotted on a brain surface. It would also be useful to compare these spatial maps to other established measures derived from the T1 scans, such as regional volume and/or cortical thickness.

20.  Results: Please report estimates of effect size for all analyses (in addition to p values).

21.  Table 3: The authors interpret the asymmetric difference in regional FD differences to reflect patterns of “lateralized atrophy”. However, the lack of symmetry / anatomical consistency might also imply that these differences are instead driven by noise, given the relatively small sample size. Moreover, the p values are relatively high and most of them are not likely to survive correction for multiple comparisons.

22.  Figure 2: To aid clarity of interpretation, please use the same color scale / range for panels A and B.

23.  What is the justification that correlating patterns of FD should be anatomically meaningful? Moreover, why would these expected to be organized according to lobes? An expected property of valid brain “networks” would be the observation of strong positive correlations within modules and weak correlations between modules. With the possible exception of the parietal lobe in controls (Figure 2A), I am not convinced that these data capture meaningful network organization.

24.  Moreover, the correlation plot in MCIs (Figure 2B) appears to be largely driven by noise. This raises the potential limitation of data quality in MCIs. As these participants are noted to have observable atrophy, this may contribute to difficulty in segmenting and parcellating the cortical surface, which may have downstream effects on the reliability of the FD estimates. What steps were taken to ensure that the MRI data and inputs for the FD calculation (particularly in the MCI sample) were of acceptable quality?

25.  Lines 243-254: The authors provide an inadequate qualitative description (more or fewer blue, green, or yellow dots) to address quantitative research questions. What were the average correlation values within and between the lobes (numeric correlation ratio and intra-lobe connectivity values)? How do those numeric values differ between MCI and controls?

26.  Line 255: Patterns of correlation among FD values do not offer any validated index of (presumably anatomical) “links” or “functional dissociation between lobes”. This statement seems to conflate the current methods with other methods to measure structural and functional brain networks with DTI and FC.

27.  Line 257: If this test was performed using the t test procedure described above, it is inappropriate. Instead, the authors should calculate the difference in Q values between groups for each of the permutations (ideally at least 1000). Then, the p value can be calculated as the proportion of permutations in which a difference as or more extreme than the observed difference was detected (see comments and links above).

28.  Figure 3: The glass brain figures are somewhat unclear to interpret. Since the data was extracted from a surface-level parcellation, information regarding module assignment would be communicated more effectively by plotting the module assignment on a brain surface (e.g., using ggseg, workbench, or any preferred means of visualizing quantitative brain data on the surface rather than volume space).

29.  Line 281: Again, it does not seem appropriate or valid to describe differences in FD-based network analyses to reflect “functional integration”.

30.  Section 3.3: What methods / sources were used to assign functional associations to ROIs?

31.  Section 3.3: How do the identified modules correspond to more established brain networks, e.g., default mode, fronto-parietal, etc.?

32.  Line 346: The authors cannot infer that patterns of correlation in FD values indicate “association fibers”.

33.  Line 352: What test was performed to determine that “MCIs showed significantly decreased intra-lobe connectivity in all cerebral lobes”?

34.  Figure 4: The presentation of left vs. right vs. interhemispheric correlations using different colors is somewhat unclear. It may be simpler to plot the left hemisphere ROIs on the left side of the circle and the right hemispheric ROIs on the right, as was done in Figure 5. Then hemispheric information can be easily interpreted among the correlations.

35.  Much of the text throughout the results is redundant, repetitive, and simply describes the figures without providing necessary quantitative analysis. For instance, descriptions of the color labeling of points should be put in the figure captions, not the text.

36.  Line 387: Again “significance” is determined by visually examining a figure without reporting results of an appropriate statistical test.

37.  Discussion: The authors are again over-interpreting their results. I see no justification that patterns of correlation between FD may reflect “commissural fibers”, “long association fibers”, “new wiring”, or “compensation”.

38.  Line 439: The authors did not compare FD to any established measure of atrophy (i.e., cortical thickness or volume) for detecting differences in MCI. They have not justification to claim that FD is “superior” in any way or that it is even a measure of “atrophy.”

References

King RD, Brown B, Hwang M, Jeon T, George AT. 2010. Fractal dimension analysis of the cortical ribbon in mild Alzheimer’s disease. Neuroimage 53:471–479. doi:10.1016/j.neuroimage.2010.06.050

Power JD, Schlaggar BL, Petersen SE. 2014. Studying Brain Organization via Spontaneous fMRI Signal. Neuron 84:681–696. doi:10.1016/j.neuron.2014.09.007

Author Response

Brainsci-2004189

Reviewer # 2

A complete, detailed list of comments is below:

  1. Abstract: what is meant by “cerebral morphological alteration in detail region”? The authors should be more explicit.

Author’s response: we thank reviewer’s suggestion. In the revision of manuscript, we have revised the original sentence “Cerebral morphological alteration in detail region can provide an accurate predictor for early recognition of MCI”. And the revised one is as “Cerebral morphological alteration in sub-regions can provide an accurate predictor for early recognition of MCI.

  1. Abstract: Line 25 is also unclear. The complexity measure is “superior” to what?

Author’s response: we thank reviewer’s comment. We have delete the word “superior” in the revised manuscript to make the sentence clearer and easy to read, and the revised sentence is as” A complexity measure termed fractal dimension(FD) was applied to assess morphological changes in cortical sub-regions of participants.”

  1. Abstract: Line 30: Please use terms like “control participants” or “cognitively normal sample”, etc., instead of simply “normal.”

Author’s response: we greatly appreciate reviewer’s valuable suggestion. Of course, the terms of “control participants” is more precisely to define the control group than using the term like “normal”. In our revised manuscript, we have used term “control participants” and “controls” to replace the term “normal group ”  in the original manuscript. The sentence in line 30 of the original abstract was as “The normal had five modules, MCI had six modules in their brain network.” And, the revised sentence is as” The controls had five modules, MCI had six modules in their brain networks.

The abstract of the revised manuscript is as follows:

  1. Line 46: MCI is typically considered a “prodromal” stage of AD. “Preclinical” often refers to cognitively normal participants with present amyloid.

Author’s response: The original sentence was as” Numerous studies have shown that MCI can be considered a preclinical stage of AD, and that MCI progresses to dementia at a rate of between 8% and 15% per year, meaning that it is an important condition to identify and treat [4].” And we had revised it as “Numerous studies have shown that MCI can be considered a prodromal stage of AD. Identify and treatment of MCI is very important because this disease progresses to dementia at a rate of between 8% and 15% per year [4].”

  1. Line 50: MCT -> MCI?

Author’s response: We appreciate reviewer’s carefully reviewed our manuscript. We had correct the typo error in sentence of line 50. The original sentence was as” In addition to cognitive decline, subjects with MCT show cortical atrophy in some specific regions.” And, the revised sentence is as “In addition to cognitive decline, MCI patients show cortical atrophy in some specific regions.”

  1. Participants: Please provide a table summarizing key demographic details in the MCI and control participants (age, sex, education, any available neuropsychological testing / cognitive screening measures).

Author’s response: We thank reviewer’s suggestions. In the revised manuscript, we have added 10 more MCI patients and 10 more healthy controls, and we also have added a table as “Table 1 “to summarize the demographics and characteristics of control participants and MCI groups in this study. The revised manuscript is as follows:

Table 1 summarizes the demographics and characteristics of control participants and MCI groups in this study. A total of 30 patients with MCI (male/female: 14/16) and 30 (male/female:15/15) healthy controls participated in this study.

Table 1 Sample demographics and characteristics of control participants and MCI groups

MCI (n=30)

CP (n=30)

p values

Age, years

70+5.2

69+3.7

0.57

Sex, female/male

16/14

15/15

0.8003

Dominant hand, right/ left

30/0

30/0

Education, years

11.3+3.6

11.4+3.4

0.558

MMSE

24.76+3.22

28.86+0.74

< 0.001

CDR global

0.5

0

CDR Memory

0.5

0

MCI, mild cognitive impairment; CP, control participants; MMSE, mini-mental state examination; CDR, clinical dementia rating scale; p value was obtained using a two-sample two-tail t test.

  1. The sample of 20 MCI and 20 controls is rather small. I see no reason why these analyses couldn’t be performed in a larger, publicly available dataset (e.g., ADNI). In fact, the authors cite a prior study that appears to have applied a similar FD analysis in ADNI (King et al., 2010). In the interest of validating these results, I would suggest replicating the analyses in a larger, independent cohort.

Author’s response: we thank reviewer’s comments and valuable suggestions. Because the founding of this manuscript is from Shin Kong Wu Ho-Su Memorial Hospital (SK Hospital), hence, all the subjects of this paper must be from Shin Kong Hospital and cannot include other non-Sin Kong Hospital participants. We apologize for having such participant request restrictions. We hope that the reviewers will understand the problems we face and the expediency of only allowing SKH patients to participate in this paper. In the revised manuscript, we had added 10 more MCI and 10 more control participants, and the new recruited subjects were diagnosed and carefully checked by clinicians. We also had added a new table as Table 1 in the revised manuscript to report the demographics and characteristics of new recruited subjects, please refer to the author’s response of comment 6.  

  1.  Line 129: Which version of FreeSurfer was implemented?

Author’s response: The version of FreeSurfer we used is Version 6. The revised manuscript is as follows: “The cortex was then parcellated and aligned into 68 sub-regions of interest (ROIs) with the Desikan–Killiany cortical atlas (DK atlas) [25] structures by using the FreeSurfer (Version 6) toolbox in MATLAB R2019b software (MathWorks, Natick, MA, USA).”

  1. Line 146: What is the justification for setting the initial value to r = 10?

Author’s response: We thank reviewer’s suggestion. Previous FD studies of cerebral complexity, all showed the proper range of box size for calculating FD value is from 2 to 8 mm. Please refer to the papers below.

  1. Line 152 and Figure 1 indicate that only a subset of the box sizes are considered when finding the best fitting regression line (i.e., the blue line and red line have different ranges). The justification and potential implications of this choice are not clear. For instance, why not fit a regression to the full range and calculate its slope? How much heterogeneity is there in the ranges of selected box sizes between different regions? Between different participants?

Author’s response: We thank reviewer’s suggestion. Please refer to the response of comment 9. In this study, we found the range of box size from 2 to 8 mm can reach the highest slope correlation.

  1. Line 163: What values are being correlated in this approach? Presumably this is referring to FD values across individual participants, but this is not clearly stated.

Author’s response: we apologize for the misunderstanding. In the revised manuscript, we have rewritten the paragraph and the revised manuscript is as follows:

“There are two steps to build a structural brain network: one is to calculate the correlation between pairs of sub-regions to indicate the strength of the inter-regional connectivity. Thus, the brain structure network is derived from the 68*68 correlation matrix of the FD values of the paired regions. Secondly, we used a modular analysis to separate the different brain distinctions into modules based on their inter-regional connections [27].”

  1. Line 163: The definition of modularity (Q) is unclear as stated. Please be more explicit and provide the equation that was used.

Author’s response: we have added a paragraph and equation in section 2.4 to explain the definition of modularity (Q). the added paragraph is as follows:

  1. Line 170: What threshold was used to define edges and modules in the current analyses is the justification of this threshold? Are the results consistent across a range of threshold values? This is of course a very common concern for network analyses.

Author’s response: we thank reviewer’s suggestion. In the revised manuscript, we had added a paragraph to explain the concept, and the added paragraph is as follows

“In this study, we have set a proportional value of 0.2 as a threshold to filter the connectivity matrix by preserving 20% proportion of the strongest correlation coefficients. In this process, all other entries below the threshold, negative correlations and all entries on the main diagonal (self-to-self connections) are set to 0 and the links will not exist. “

  1. Line 190: “A smaller number of participation coefficient indicates a large degree of intra-modal connectivity within node i”. Is this true? Shouldn’t a smaller participation coefficient simply indicate a smaller degree of inter-modal connectivity at node i?

Author’s response: we thank reviewer’s comment. In the revised manuscript, we have rewritten the paragraph and the revised paragraph is as follows:

  1. Statistical analyses: The authors are testing for regional differences in FD in 68 ROIs, and thus increase the type 1 error rate. It is necessary to apply appropriate correction for multiple comparisons.

Author’s response: we have rewritten the paragraph, and performed FDR and Effect Size test. The revised paragraph is as follows:

“2.5. Statistical Analysis

In this study, a two-tailed t-test and multiple false discovery rate (FDR) correction were used to compare measurements [29], including FD values and brain structural network parameters, between the control and MCI patient groups. Then, the effect size process was applied to measure the strength of the relationship between compared variables of groups to indicate the practical difference [30]. Note that each group with 30 subjects had 68 FD values for each region. We computed the FD value on the basis of correlation between any two regions. As a result, a 68 × 68 correlation map was obtained for each group to build a structural network, resulting in one set of topological properties for each structural network. Accordingly, we could not directly perform any statistical comparison of the corresponding topological properties between these two structural networks.

  1. Dey, Soumen & Delampady, Mohan. (2013). False discovery rates and multiple testing. Resonance. 18. 10.1007/s12045-013-0137-9.
  2. Kelley, Ken; Preacher, Kristopher J. (2012). "On Effect Size". Psychological Methods17(2): 137–152.

  1. Line 209: For the permutation test, why were only 10 MCI participants and 10 controls reassigned? What happens to the other 10 participants in each group - are they fixed across permutations? If so, this is an inappropriate implementation. If not, I see no reason why the full sample of 20 MCI and 20 controls shouldn’t be randomly reassigned, as is done in a typical permutation / randomization test. Further 100 permutations seems a little small. I would suggest at least 1000. Finally, this paragraph seems to contain several redundant sentences.

Author’s response: in each permutation, we randomly selected 10MCI and 10 controls to perform network analysis. We performed the process 1000 times, and got 1000 network parameters to compare the difference between groups. The revised manuscript is as follows:

“In this study, a permutation test was conducted to statistically compare the differences in network properties between the two groups [30]. To test the null hypothesis, we randomly selected 10 MCI and 10 controls from each study group (30 subjects) and reassigned these 20 subjects as the randomized MCI group and randomized control group, separately. This randomized simulation and recalculation of the network properties were repeated 1000 times to compute the correlation matrix for each randomized group. The 95th percentile points of each distribution of the 1000 simulations were used as critical values in a two-sample, one-tailed t test to reject the null hypothesis, with a type I error probability of 0.05. Then, the network properties Q, P, and Z were calculated for each reassigned correlation matrix of the two groups. Following the permutation process, 1000 sets of network parameters were used in a two-sample, one-tailed t test with FDR correction to assess significant differences between the study groups.”

  1. Line 215: It is inappropriate to run a t test on bootstrapped permutations. This creates a problem such that non-meaningful differences would be considered significant simply by adding more and more permutations. Instead, the appropriate test is to generate an “empirical p value” by calculating the proportion of permutations in which an effect (i.e., difference between group 1 and group 2) as or more extreme than the observed difference. For demonstration, see, e.g.: https://rpubs.com/valentin/permutation-test-1 or https://towardsdatascience.com/how-to-use-permutation-tests-bacc79f45749

Author’s response: please refer to the author’s response of comment 17.

  1. Line 218: The authors interpret reductions in FD values to reflect “atrophy”. What is the justification for this interpretation? How do these measures compare to more established indicators of “atrophy”, namely regional volume and cortical thickness?

Author’s response: In the revised manuscript, we have revised the sub-title of section 3.1 as: 3.1. Patients with MCI exhibited significant lateralized FD changes mainly in temporal lobes regions. Previous FD studies have reported that FD are more suitable to detect cerebral atrophy. The interpretation of reductions in FD values to reflect “atrophy” can be found from our previous study. Please refer the reference below.

This study employs 3D FD analysis to quantify the degeneration of the CBWM and CBGM in MSA-C patients. Fig. 7 shows that because the FD is an index of morphological complexity (Esteban et al., 2007), a higher FD value indicates a more complex cerebellar structure, while a decrease in the cerebellar FD value may indicate a degeneration of the cerebellar structure. Since the 3D CB of MSA-C patients manifested less folding patterns and a smaller volumetric size, smaller FD values and volumetric values were anticipated.

The revised section 3.1 ia as follows:

  1. Results

3.1. Patients with MCI exhibited significant lateralized FD changes mainly in temporal lobes regions

Table 3 summarizes the FD values of each lobe in control and MCI groups. The MCI group revealed significantly lower FD values in their bilateral temporal lobes (left temporal: p<0.001, right temporal: p=0.0025) and right parietal lobe(p=0.0015).

Table 3. FD measure results of cerebral lobes in Control and MCI groups.

Lobe

Control

MCI

P value

Effect size

Frontal(L)

2.2324+0.0161

2.2319+0.0224

0.9149

0.0128

Frontal(R)

2.2393+0.0144

2.2331+0.0213

0.1858

0.1681

Temporal (L)

2.2108+0.0108

2.1908+0.02

<0.001*

0.5283

Temporal (R)

2.1985+0.0146

2.1849+0.0189

0.0025*

0.3735

Parietal(L)

2.2902+0.0183

2.2844+0.0158

0.1922

0.1672

Parietal(R)

2.2990+0.0184

2.2822+0.0209

0.0015*

0.3924

Occipital(L)

2.2146+0.0301

2.2084+0.0299

0.4262

0.1028

Occipital(R)

2.2247+0.0334

2.2159+0.024

0.2469

0.1498

  1. Results: It would be very helpful to report the regional FD values visually plotted on a brain surface. It would also be useful to compare these spatial maps to other established measures derived from the T1 scans, such as regional volume and/or cortical thickness.

Author’s response: we thank reviewer’s suggestion. Because some cortical parcellated regions such as cingulate, insula, para hippocampal and so on are located beneath the cortical surface. Sorry, hence we used a brain volume space in this study. We will take the second suggestion of compare FD measure with volume measure or cortical thickness in our future studies.

  1. Results: Please report estimates of effect size for all analyses (in addition to p values).

Author’s response: we have rewritten the statistical analysis section paragraph, and performed FDR and Effect Size test. The revised paragraph is as follows:

“2.5. Statistical Analysis

In this study, a two-tailed t-test and multiple false discovery rate (FDR) correction were used to compare measurements [29], including FD values and brain structural network parameters, between the control and MCI patient groups. Then, the effect size process was applied to measure the strength of the relationship between compared variables of groups to indicate the practical difference [30]. Note that each group with 30 subjects had 68 FD values for each region. We computed the FD value on the basis of correlation between any two regions. As a result, a 68 × 68 correlation map was obtained for each group to build a structural network, resulting in one set of topological properties for each structural network. Accordingly, we could not directly perform any statistical comparison of the corresponding topological properties between these two structural networks.

3.1. Patients with MCI exhibited significant lateralized atrophy mainly in temporal lobes regions

Table 3 summarizes the FD values of each lobe in control and MCI groups. The MCI group revealed significantly lower FD values in their bilateral temporal lobes (left temporal: p<0.001, right temporal: p=0.0025) and right parietal lobe(p=0.0015).

Table 3. FD measure results of cerebral lobes in Control and MCI groups.

Lobe

Control

MCI

P value

Effect size

Frontal(L)

2.2324+0.0161

2.2319+0.0224

0.9149

0.0128

Frontal(R)

2.2393+0.0144

2.2331+0.0213

0.1858

0.1681

Temporal (L)

2.2108+0.0108

2.1908+0.02

<0.001*

0.5283

Temporal (R)

2.1985+0.0146

2.1849+0.0189

0.0025*

0.3735

Parietal(L)

2.2902+0.0183

2.2844+0.0158

0.1922

0.1672

Parietal(R)

2.2990+0.0184

2.2822+0.0209

0.0015*

0.3924

Occipital(L)

2.2146+0.0301

2.2084+0.0299

0.4262

0.1028

Occipital(R)

2.2247+0.0334

2.2159+0.024

0.2469

0.1498

Table 4 summarizes the sub-regions with significantly decreased FD values in MCI. MCI showed 27 sub-regions with significantly decreased FD values, and mainly in the right hemisphere (L/R:10/17). Of these 27 significantly decreased FD sub-regions, 11 were in the temporal lobe (L/R:6/5), 7 in the frontal lobe (L/R:2/5), 7 in the parietal lobe(L/R:2/5) and 2 in the right occipital lobe.

Table 4. Cortical sub-regions of significant decreased FD values in MCI group.

Frontal(L)

Left hemisphere

controls

MCI

P value

Effect size

Paracentral

2.1765+0.0523

2.1546+0.0443

0.0268

0.22

Rostral middle frontal

2.4095+0.0178

2.4009+0.0247

0.0353

0.20

Frontal(R)

Caudal anterior cingulate

2.1322+0.0554

2.1037+0.0497

0.0196

0.26

Caudal middle frontal

2.2911+0.0448

2.2766+0.0335

0.0363

0.18

Medial Orbital Frontal 

2.2473+0.0359

2.2235+0.0776

0.0342

0.19

Paracentral

2.2118+0.0361

2.1724+0.0378

0.0003

0.47

Superior frontal

2.4082+0.0169

2.3988+0.0166

0.0189

0.27

Temporal(L)

Medial Temporal          

2.3148+0.0274

2.2970+0.0334

0.0245

0.28

Fusiform

2.2993+0.0271

2.2853+0.0248

0.0183

0.26

Inferior temporal

2.3358+0.020

2.3223+0.0294

0.0182

0.26

Transverse Temporal

2.0486+0.0498

2.0229+0.0595

0.0248

0.23

Entorhinal

2.1278+0.0415

2.0766+0.0518

0.0004

0.47

Temporal pole

2.1937+0.0352

2.1734+0.0445

0.0213

0.25

Temporal(R)

Bankssts   

2.1815+0.0385

2.1622+0.0562

0.0352

0.20

Fusiform    

2.2994+0.0268

2.2816+0.0341

0.0195

0.28

Para hippocampal   

2.0464+0.0590

2.0165+0.050

0.0200

0.26

Transverse Temporal    

1.9913+0.0621

1.9542+0.0654

0.0216

0.28

Superior temporal

2.3435+0.0351

2.333+0.0281

0.0438

0.16

Parietal(L)

  Postcentral

2.2919+0.02879

2.2807+0.0267

0.0368

0.20

Supra marginal

2.3867+0.0271

2.3737+0.0263

0.0242

0.28

Parietal(R)

Inferior parietal

2.4292+0.020

2.4181+0.0171

0.0275

0.27

Superior parietal

2.3581+0.0221

2.3424+0.0193

0.0076

0.35

Postcentral

2.2829+0.02169

2.2624+0.0279

0.0047

0.38

Poster cingulate

2.2097+0.0602

2.1864+0.0652

0.0362

0.18

Supra marginal

2.3886+0.02437

2.3574+0.0329

0.0342

0.47

Occipital(R)

Lateral occipital

2.3713+0.0256

2.3619+0.0261

0.0357

0.18

lingual

2.2721+0.0257

2.2528+0.0373

0.0288

0.29

     In result section 3.3

“After 1000 permutations were calculated, the network modularity value (Q) was significantly lower in the MCI group than in the control group (normal: 0.2548+0.0057, MCI: 0.2451+0.0066, P < 0.05, Effect Size=0.62).”

The effect size calculators

https://lbecker.uccs.edu/#Calculate%20d%20and%20r%20usaaing%20t%20values%20(separate%20groups)

  1. Table 3: The authors interpret the asymmetric difference in regional FD differences to reflect patterns of “lateralized atrophy”. However, the lack of symmetry / anatomical consistency might also imply that these differences are instead driven by noise, given the relatively small sample size. Moreover, the p values are relatively high and most of them are not likely to survive correction for multiple comparisons.

Author’s response: we thank reviewer’s suggestion, the original Table 3 had been revised as Table 4 in the revised manuscript and had added the effect size value please in Table 4, please refer to response for comment 20.

  1. Figure 2: To aid clarity of interpretation, please use the same color scale / range for panels A and B.

Author’s response: we greatly appreciate reviewer’s suggestion. The revised figure 2 is as follows:

       Figure 2. Sub-region correlation maps between different brain lobes in the control group (Figure 2(a)) and MCI group (Figure 2(b)). There are 68 rows and 68 columns in each plot, and the dots from the first row and first column from the top left indicate the correlation rate of the first sub-region of the ROI (left anterior cingulate) with the other 67 sub-regions, the second row and second column from the top left indicate the correlation rate of the second sub-region of the ROI (right anterior cingulate) with the other 67 sub-regions, and so on. In each figure, sub-regions of the frontal lobe are labeled within the red line, the temporal lobe is labeled within the purple line, the parietal lobe is labeled within the white line, and the occipital lobe is labeled within the green line. The color bars indicate the density of correlation, and the color bars of the control group are scaled higher than those of the MCI group. For each brain lobe, normal controls showed higher correlation densities within the lobes and with other lobes than the MCI group.

  1. What is the justification that correlating patterns of FD should be anatomically meaningful? Moreover, why would these expected to be organized according to lobes? An expected property of valid brain “networks” would be the observation of strong positive correlations within modules and weak correlations between modules. With the possible exception of the parietal lobe in controls (Figure 2A), I am not convinced that these data capture meaningful network organization.

Author’s response: In brain structure networks, regional brain thickness or volume is used as a nodal factor to calculate the correlation between nodes. Nodes with high correlation are edges and groups of nodes with high correlation are modules.  In this study, we use the intra-modular connectivity(Zi) of each sub-region within lobe as the nodes in a module to calculate the intra-module connectivity. The intra-lobe connectivity was driven from network analysis, not directly from figure 2A or figure 2B.

  1. Moreover, the correlation plot in MCIs (Figure 2B) appears to be largely driven by noise. This raises the potential limitation of data quality in MCIs. As these participants are noted to have observable atrophy, this may contribute to difficulty in segmenting and parcellating the cortical surface, which may have downstream effects on the reliability of the FD estimates. What steps were taken to ensure that the MRI data and inputs for the FD calculation (particularly in the MCI sample) were of acceptable quality?

Author’s response: The thickness of the MRI slice is very important for FD measurements. If the thickness of the slice is greater than the size of the box used to calculate the FD, then it is not possible to obtain a correct FD value. The thickness of the slice must be smaller than the size of the counting box. In this study, the minimum size of the counting box was 2 mm, while the thickness of the MRI slice was 1.5 mm. Therefore, in this study, the FD calculation was appropriate for patients with MCI and had an accurate result.

  1. Lines 243-254: The authors provide an inadequate qualitative description (more or fewer blue, green, or yellow dots) to address quantitative research questions. What were the average correlation values within and between the lobes (numeric correlation ratio and intra-lobe connectivity values)? How do those numeric values differ between MCI and controls?

Author’s response: we greatly appreciate reviewer’s suggestion. In the revised manuscript, we have rephrased section 3.2 and added the average correlation values within and between the lobes as the reviewer’s suggestion. The revised manuscript is as follows:

“ Figure 2(a) and (b) illustrate the correlation map of sub-regions between different lobes of control and MCI groups. The color bar indicates the correlation rate. Firstly, the control participants group shows a higher range of correlation rate from zero to 0.8, while the MCI group shows a lower correlation rate from zero to 0.65. For controls, the mean correlate rate of frontal lobe is 0.3276, 0.2754 for the temporal lobe, 0.3719 for the parietal lobe, and 0.2888 for the occipital lobe. The MCI revealed a lower correlation rate in each lobe. They sowed the correlate rate of 0.1679 for the frontal lobe, 0.1861 for the temporal lobe, 0.1451 for the parietal lobe, and 0.1540 for the occipital lobe.”

  1. Line 255: Patterns of correlation among FD values do not offer any validated index of (presumably anatomical) “links” or “functional dissociation between lobes”. This statement seems to conflate the current methods with other methods to measure structural and functional brain networks with DTI and FC.

Author’s response: the revised manuscript is as follows:

” Figure 2(a) and (b) illustrate the correlation map of sub-regions between different lobes of control and MCI groups. The color bar indicates the correlation rate. Firstly, the control participants group shows a higher range of correlation rate from zero to 0.8, while the MCI group shows a lower correlation rate from zero to 0.65. For controls, the mean correlate rate of frontal lobe is 0.3276, 0.2754 for the temporal lobe, 0.3719 for the parietal lobe, and 0.2888 for the occipital lobe. The MCI revealed a lower correlation rate in each lobe. They sowed the correlate rate of 0.1679 for the frontal lobe, 0.1861 for the temporal lobe, 0.1451 for the parietal lobe, and 0.1540 for the occipital lobe.”

2.4. Network property analysis of intra-modular and inter-modular connectivity

There are two steps to build a structural brain network: one is to calculate the correlation between pairs of sub-regions to indicate the strength of the inter-regional connectivity. Thus, the brain structure network is derived from the 68*68 correlation matrix of the FD values of the paired regions. Secondly, we used a modular analysis to separate the different brain distinctions into modules based on their inter-regional connections [27].  Modularity (Q) indicates the number of edges where all pairs of nodes within the same module.

The Q can be expressed as:

where A is the connection matrix of the network, and each element of A is the correlation coefficient between regions;  is defined as the sum of the correlation coefficient between node i and its connected regions, and is also called the degree of node I;  represents the total number of edges; and  denotes the module of node i. The  -function  is 1 when nodes i and j belong to the same module and 0 otherwise. If the investigated network presents superior partitioning, it will have a greater Q value and is more likely to construct a modular organization [21].

In this study, we have set a proportional value of 0.2 as a threshold to filter the connectivity matrix by preserving 20% proportion of the strongest correlation coefficients. In this process, all other entries below the threshold, negative correlations and all entries on the main diagonal (self-to-self connections) are set to 0 and the links will not exist.”

  1. Line 257: If this test was performed using the t test procedure described above, it is inappropriate. Instead, the authors should calculate the difference in Q values between groups for each of the permutations (ideally at least 1000). Then, the p value can be calculated as the proportion of permutations in which a difference as or more extreme than the observed difference was detected (see comments and links above).

Author’s response: In the revised manuscript, we have performed 1000 times of permutation and have 1000 Q values for MCI and control groups. We compare the 1000 Q values between MCI and controls with FDR correction and take the mean of the 1000 p values.

“2.5. Statistical Analysis

In this study, a two-tailed t-test and multiple false discovery rate (FDR) correction were used to compare measurements [29], including FD values and brain structural network parameters, between the control and MCI patient groups. Then, the effect size process was applied to measure the strength of the relationship between compared variables of groups to indicate the practical difference [30]. Note that each group with 30 subjects had 68 FD values for each region. We computed the FD value on the basis of correlation between any two regions. As a result, a 68 × 68 correlation map was obtained for each group to build a structural network, resulting in one set of topological properties for each structural network. Accordingly, we could not directly perform any statistical comparison of the corresponding topological properties between these two structural networks.

In this study, a permutation test was conducted to statistically compare the differences in network properties between the two groups [31]. To test the null hypothesis, we randomly selected 10 MCI and 10 controls from each study group (30 subjects) and reassigned these 20 subjects as the randomized MCI group and randomized control group, separately. This randomized simulation and recalculation of the network properties were repeated 1000 times to compute the correlation matrix for each randomized group. The 95th percentile points of each distribution of the 1000 simulations were used as critical values in a two-sample, one-tailed t test to reject the null hypothesis, with a type I error probability of 0.05. Then, the network properties Q, P, and Z were calculated for each reassigned correlation matrix of the two groups. Following the permutation process, 1000 sets of network parameters were used in a two-sample, one-tailed t test with FDR correction to assess significant differences between the study groups. “

  1. 2Figure 3: The glass brain figures are somewhat unclear to interpret. Since the data was extracted from a surface-level parcellation, information regarding module assignment would be communicated more effectively by plotting the module assignment on a brain surface (e.g., using ggseg, workbench, or any preferred means of visualizing quantitative brain data on the surface rather than volume space).

Author’s response: we greatly thank reviewer’s suggestion. Because some cortical parcellated regions such as cingulate, insula, para hippocampal and so on are located beneath the cortical surface. Sorry, hence we used a brain volume space to illustrate the distribution nodes of modules.

  1. Line 281: Again, it does not seem appropriate or valid to describe differences in FD-based network analyses to reflect “functional integration”.

Author’s response: we have deleted the sentence and revised the paragraph as follows:

“3.2. Patients with MCI exhibited lower correlation rates within and between lobes

Figures 2(a) and (b) illustrate the correlation plots between different brain lobes in the control and MCI groups for the sub-regions. The color bars indicate the correlation rates. First, the control group shows a higher correlation rate range from 0 to 0.8, while the MCI group shows a lower correlation rate from 0 to 0.65. For the control group, the mean correlation rate is 0.3276 for the frontal lobe, 0.2754 for the temporal lobe, 0.3719 for the parietal lobe, and 0.2888 for the occipital lobe. MCI shows lower correlation rates for each lobe. They showed correlation rates of 0.1679 for the frontal lobe, 0.1861 for the temporal lobe, 0.1451 for the parietal lobe, and 0.1540 for the occipital lobe.”

  1. Section 3.3: What methods / sources were used to assign functional associations to ROIs?

Author’s response: In the revised version, we removed the sentences associated with these functions from the original draft. We have rewritten and shortened the Result section 3.3, and the revised section 3.3 reads as follows.

“3.3. MCI revealed smaller modular size and less nodes integration in their brain structural network

Figures 3(a) and (b) illustrate the node distribution of modules in the FD-based brain structure network for the control and MCI groups, respectively. The figures were plotted using BrainNet Viewer software [32]. According to the network analysis, the 68 sub-regions of the cortex in the control group were clustered into five middle segments, whereas the 68 sub-regions of the cortex in MCI were clustered into six modules.

Table 5(a) and Table 5(b) summarize the detailed sub-regions within each module of the brain module network for the normal and MCI groups, respectively. In each module, the top row indicates nodes in the left hemisphere and the bottom row indicates nodes in the right hemisphere. In this study, we defined the largest module as the first module, the second largest module as the second module, and so on. The control group had three larger modules (Module 1, 19 nodes; Module 2, 17 nodes; Module 3, 17 nodes), and two smaller modules (Module 4, 8 nodes; Module 5, 7 nodes). The MCI showed smaller module sizes in their brain network than the control group. There was only one larger module (module 1, 18 nodes), three medium-sized modules (module 2, 13 nodes; module 3, 11 nodes; module 4, 10 nodes), and two smaller modules (module 5, 9 nodes; module 6, 7 nodes) in MCI's brain network. After 1000 permutations were calculated, the network modularity value (Q) was significantly lower in the MCI group than in the control group (normal: 2548, MCI: 0.2451, P < 0.05, FDR corrected). This result implies a relatively low density and efficiency of the structural brain network in MCI.

For the control group, modules 1, 2 and 3 all show the integration of nodes from the four functional lobes (frontal, temporal, parietal and occipital). In module 1, there were three pairs of bilateral nodes and in module 2, there were four pairs of bilateral nodes (bolded in Table 5(a)). Unlike controls, MCI showed sparse clustering and less functional node integration in its brain structural network module groupings. In MCI's modules 1, 2, and 3, nodes in the frontal, parietal, and temporal lobes were integrated, but none of the nodes in the occipital lobe. MCI also showed fewer bilateral node pairs in its brain structural network modules. In Module 1, there are two bilateral node pairs linked. In modules 2 and 3 of MCI, there are only one pair of bilateral node links (bolded in Table 5(b)).”

  1. Section 3.3: How do the identified modules correspond to more established brain networks, e.g., default mode, fronto-parietal, etc.?

Author’s response: We have added a paragraph to the Discussion section to explain the correspondence of our results with other functional network studies, and the added paragraph in the discussion section is as follows

“4.2. MCI show shrinkage modular size and less functional lobes integration in their brain structural network

 Structural networks are believed to shape and provide constraints for the dynamics of functional connectivity, to some extent, it has been widely acknowledged that functional networks can be predicted from the underlying structural connectome [39]. A high goodness of fit level for the structure-function mapping of brain networks has been reported [40,41], as well as a pattern dependence between the connection matrices of the resting-state functional and structural networks [41]. Robust model analysis also reflects a reliable combination of structural and functional networks that are optimally correlated, with the structural network predicting the functional network, but the two networks not necessarily overlapping [40].

Functional network studies have reported that cerebral sub-regions that exhibit different combinations of control signals in many tasks can be grouped into three distinct networks, namely the Fronto-Parietal network (FPN), Cingulo-Opercular network (CON) and default model network (DMN) [42-44]. The FPN includes the prefrontal, middle cingulate gyrus, inferior parietal and precuneus, the CON includes the prefrontal, insula, anterior cingulate and superior frontal lobes, and the DMN includes the inferior temporal, para hippocampal, lateral parietal and posterior cingulate gyrus [42].

In the present study, we investigated and compared the brain structural network patterns in normal controls and MCI group. In the control group, we found network model 1 included many of the nodes of the FPN (frontal, parietal and precuneus). Network model 2 included many of the nodes of the DMN (para hippocampal (Left and right) and posterior cingulate (left and right)). Network model 3 included many nodes of the CON (anterior cingulate, superior frontal and insula).”

The added references are as follow:

  1. O’Neill, G.C., Tewarie, P., Vidaurre, D., Liuzzi, L., Woolrich, M.W., Brookes, M.J., 2017. Dynamics of large-scale electrophysiological networks: a technical review. Neuroimage 180, part B, 559–576.
  2. Miši´c, B.; Betzel, R.F.; De Reus, M.A.; Van Den Heuvel, M.P.; Berman, M.G.; McIntosh, A.R.; Sporns, O. Network-level structure-function relationships in human neocortex. Cereb. Cortex 2016, 26, 3285–3296.
  3. Meier J, Tewarie P, Hillebrand A, Douw L, van Dijk BW, Stufflebeam SM, Van Mieghem P. A Mapping Between Structural and Functional Brain Networks. Brain Connect. 2016 May;6(4):298-311. doi: 10.1089/brain.2015.0408. Epub 2016 Mar 29. PMID: 26860437; PMCID: PMC4939447.
  4. Fair DA, Cohen AL, Power JD, Dosenbach NUF, Church JA, et al. (2009) Functional Brain Networks Develop from a ‘‘Local to Distributed’’ Organization. PLoS Comput Biol 5(5): e1000381. doi:10.1371/journal.pcbi.1000381
  5. Dosenbach NU, Fair DA, Miezin FM, Cohen AL, Wenger KK, et al. (2007) Distinct brain networks for adaptive and stable task control in humans. Proc Natl Acad Sci U S A 104: 11073–11078.
  6. Dosenbach NU, Visscher KM, Palmer ED, Miezin FM, Wenger KK, et al. (2006) A core system for the implementation of task sets. Neuron 50: 799–812.
  7. Line 346: The authors cannot infer that patterns of correlation in FD values indicate “association fibers”.

 Author’s response: Sorry for the misuse error. In the revised manuscript, we have deleted all the words “association fiber”. In the revised manuscript, we have rephrased and shortened the results section. In the original manuscript, line 346 was in results section 3.4. The revised section 3.4 is as follows

“3.4. MCI revealed significant alteration of intra-lobe and inter-lobes connectivity in their brain structural network

Figures 4(a) to (h) illustrate the detailed connections within each brain lobe in the control and MCI groups, and Table 6 summarizes the number of connections within the left hemisphere, left hemisphere and between hemispheres. First, the link distribution in each lobe of MCI showed a similar pattern to that of the control group. In each brain lobe, MCI showed a smaller number of links and thinner link widths. The MCI group showed the most lateral link loss in the left temporal lobe (controls: 9 links, MCI: 4 links) and also the most bilateral link loss in their temporal lobe (controls: 24 links, MCI: 14 links). Compared with Figure 4(c) and (g), only two red lines in Figure 4(g) are wider than those in Figure 4(c), which may imply that the MCI group has the most severe decrease in bilateral link strength in the parietal lobe.”

  1. Line 352: What test was performed to determine that “MCIs showed significantly decreased intra-lobe connectivity in all cerebral lobes”?

Author’s response: we thank reviewer’s suggestion. In the revised manuscript, we have added a table as Table 5 to indicate the connection number within lateral lobe and between bilateral lobes. We also have rearranged and relabeled the right (R) and left (L) lobes in Figure 4 to make it more easily to read. The revised manuscript is as follows:

3.4. MCI revealed significant alteration of intra-lobe and inter-lobes connectivity in their brain structural network

Figures 4(a) to (h) illustrate the detailed connections within each brain lobe in the control and MCI groups, and Table 6 summarizes the number of connections within the left hemisphere, left hemisphere and between hemispheres. First, the link distribution in each lobe of MCI showed a similar pattern to that of the control group. In each brain lobe, MCI showed a smaller number of links and thinner link widths. The MCI group showed the most lateral link loss in the left temporal lobe (controls: 9 links, MCI: 4 links) and also the most bilateral link loss in their temporal lobe (controls: 24 links, MCI: 14 links). Compared with Figure 4(c) and (g), only two red lines in Figure 4(g) are wider than those in Figure 4(c), which may imply that the MCI group has the most severe decrease in bilateral link strength in the parietal lobe.

  1. Figure 4: The presentation of left vs. right vs. interhemispheric correlations using different colors is somewhat unclear. It may be simpler to plot the left hemisphere ROIs on the left side of the circle and the right hemispheric ROIs on the right, as was done in Figure 5. Then hemispheric information can be easily interpreted among the correlations.

Author’s response: please refer to the author’s response of comment 33.

  1. Much of the text throughout the results is redundant, repetitive, and simply describes the figures without providing necessary quantitative analysis. For instance, descriptions of the color labeling of points should be put in the figure captions, not the text.

Author’s response: We have rewritten and shorten the result section. We have added more quantitative analysis in the revised manuscript. We also have replaced the descriptions of the color labeling of points in the figure captions. Please refer to the revised manuscript.

  1. Line 387: Again “significance” is determined by visually examining a figure without reporting results of an appropriate statistical test.

Author’s response: We have deleted these sentences in the revised manuscript. We also we improve more appropriate statistical test as the reviewer had suggested, and have improved the original manuscript more organized and easy to read, thanks!

  1. Discussion: The authors are again over-interpreting their results. I see no justification that patterns of correlation between FD may reflect “commissural fibers”, “long association fibers”, “new wiring”, or “compensation”.

Author’s response: We have deleted all these words “commissural fibers”, “long association fibers”, “new wiring”, or “compensation” in the revised manuscript.

  1. Line 439: The authors did not compare FD to any established measure of atrophy (i.e., cortical thickness or volume) for detecting differences in MCI. They have not justification to claim that FD is “superior” in any way or that it is even a measure of “atrophy.”

Author’s response: we thank reviewer’s suggestion. In the revised manuscript, we have added a paragraph to compare our results with other measure of atrophy in MCI. The added paragraph in the revised discussion section 4.1 is as follows:

“4.1. FD measure revealed ability for detecting of cerebral changes in MCI

The FD approach is a consistent and most frequently chosen feature that has been proposed to calculate the intrinsic structural complexity of the cerebral cortex to predict cognitive decline in disease and can complement standard imaging [33]. Traditional methods such as cortical thickness or volume show that MCI may atrophy their cerebral cortex mainly in the medial temporal, hippocampus, entorhinal, and some sporadic reports in the para hippocampus, amygdala, fusiform gyrus, lateral temporal, parietal, frontal and occipital lobes [6-8]. However, in neurodegenerative diseases, the complexity of assessing cortical shape may better reflect symptoms of atrophy than using traditional volumetric measures [34]. In this study, we prospectively applied FD to measure cortical DK sub-regions in MCI, and the regions of atrophy we identified included those measured by conventional methods in previous studies [6-8], as well as additional sub-regions in the medial orbital frontal, paracentral, inferior parietal, and superior parietal lobes. Our results showed that the medial temporal, para hippocampal, paracentral, entorhinal, fusiform, postcentral and superior parietal were the sub-regions with more decreased FD values in MCI.

Using the same FD analysis, Nicolas Nicastro et al. reported that the orbitofrontal cortex and paracentral gyrus are particularly vulnerable in terms of memory and language impairment, and that the FD represents a sensitive imaging marker for prevention and diagnostic strategies [34]. In subjects with MCI, precise measurement of medial temporal lobe atrophy (MTA) may improve predictive accuracy and reduce false-negative classification of dementia [35]. Furthermore, it has been highlighted that visual assessment of MTA on brain MRI using a standardized rating scale is a strong and independent predictor of conversion to dementia in relatively young MCI patients [36]. With increasing duration of MCI, measuring hippocampal atrophy in older MCI patients has been reported to predict subsequent conversion to AD [37]. Structural abnormalities in the orbitofrontal cortex (OFC) may reflect a potential neurodevelopmental risk marker for MCI [38]. Taken together, our results support these previous findings in MCI and may provide a new approach for identifying MCI.”

Brainsci-2004189

Reviewer # 2

A complete, detailed list of comments is below:

  1. Abstract: what is meant by “cerebral morphological alteration in detail region”? The authors should be more explicit.

Author’s response: we thank reviewer’s suggestion. In the revision of manuscript, we have revised the original sentence “Cerebral morphological alteration in detail region can provide an accurate predictor for early recognition of MCI”. And the revised one is as “Cerebral morphological alteration in sub-regions can provide an accurate predictor for early recognition of MCI.

  1. Abstract: Line 25 is also unclear. The complexity measure is “superior” to what?

Author’s response: we thank reviewer’s comment. We have delete the word “superior” in the revised manuscript to make the sentence clearer and easy to read, and the revised sentence is as” A complexity measure termed fractal dimension(FD) was applied to assess morphological changes in cortical sub-regions of participants.”

  1. Abstract: Line 30: Please use terms like “control participants” or “cognitively normal sample”, etc., instead of simply “normal.”

Author’s response: we greatly appreciate reviewer’s valuable suggestion. Of course, the terms of “control participants” is more precisely to define the control group than using the term like “normal”. In our revised manuscript, we have used term “control participants” and “controls” to replace the term “normal group ”  in the original manuscript. The sentence in line 30 of the original abstract was as “The normal had five modules, MCI had six modules in their brain network.” And, the revised sentence is as” The controls had five modules, MCI had six modules in their brain networks.

The abstract of the revised manuscript is as follows:

  1. Line 46: MCI is typically considered a “prodromal” stage of AD. “Preclinical” often refers to cognitively normal participants with present amyloid.

Author’s response: The original sentence was as” Numerous studies have shown that MCI can be considered a preclinical stage of AD, and that MCI progresses to dementia at a rate of between 8% and 15% per year, meaning that it is an important condition to identify and treat [4].” And we had revised it as “Numerous studies have shown that MCI can be considered a prodromal stage of AD. Identify and treatment of MCI is very important because this disease progresses to dementia at a rate of between 8% and 15% per year [4].”

  1. Line 50: MCT -> MCI?

Author’s response: We appreciate reviewer’s carefully reviewed our manuscript. We had correct the typo error in sentence of line 50. The original sentence was as” In addition to cognitive decline, subjects with MCT show cortical atrophy in some specific regions.” And, the revised sentence is as “In addition to cognitive decline, MCI patients show cortical atrophy in some specific regions.”

  1. Participants: Please provide a table summarizing key demographic details in the MCI and control participants (age, sex, education, any available neuropsychological testing / cognitive screening measures).

Author’s response: We thank reviewer’s suggestions. In the revised manuscript, we have added 10 more MCI patients and 10 more healthy controls, and we also have added a table as “Table 1 “to summarize the demographics and characteristics of control participants and MCI groups in this study. The revised manuscript is as follows:

Table 1 summarizes the demographics and characteristics of control participants and MCI groups in this study. A total of 30 patients with MCI (male/female: 14/16) and 30 (male/female:15/15) healthy controls participated in this study.

Table 1 Sample demographics and characteristics of control participants and MCI groups

MCI (n=30)

CP (n=30)

p values

Age, years

70+5.2

69+3.7

0.57

Sex, female/male

16/14

15/15

0.8003

Dominant hand, right/ left

30/0

30/0

Education, years

11.3+3.6

11.4+3.4

0.558

MMSE

24.76+3.22

28.86+0.74

< 0.001

CDR global

0.5

0

CDR Memory

0.5

0

MCI, mild cognitive impairment; CP, control participants; MMSE, mini-mental state examination; CDR, clinical dementia rating scale; p value was obtained using a two-sample two-tail t test.

  1. The sample of 20 MCI and 20 controls is rather small. I see no reason why these analyses couldn’t be performed in a larger, publicly available dataset (e.g., ADNI). In fact, the authors cite a prior study that appears to have applied a similar FD analysis in ADNI (King et al., 2010). In the interest of validating these results, I would suggest replicating the analyses in a larger, independent cohort.

Author’s response: we thank reviewer’s comments and valuable suggestions. Because the founding of this manuscript is from Shin Kong Wu Ho-Su Memorial Hospital (SK Hospital), hence, all the subjects of this paper must be from Shin Kong Hospital and cannot include other non-Sin Kong Hospital participants. We apologize for having such participant request restrictions. We hope that the reviewers will understand the problems we face and the expediency of only allowing SKH patients to participate in this paper. In the revised manuscript, we had added 10 more MCI and 10 more control participants, and the new recruited subjects were diagnosed and carefully checked by clinicians. We also had added a new table as Table 1 in the revised manuscript to report the demographics and characteristics of new recruited subjects, please refer to the author’s response of comment 6.  

  1.  Line 129: Which version of FreeSurfer was implemented?

Author’s response: The version of FreeSurfer we used is Version 6. The revised manuscript is as follows: “The cortex was then parcellated and aligned into 68 sub-regions of interest (ROIs) with the Desikan–Killiany cortical atlas (DK atlas) [25] structures by using the FreeSurfer (Version 6) toolbox in MATLAB R2019b software (MathWorks, Natick, MA, USA).”

  1. Line 146: What is the justification for setting the initial value to r = 10?

Author’s response: We thank reviewer’s suggestion. Previous FD studies of cerebral complexity, all showed the proper range of box size for calculating FD value is from 2 to 8 mm. Please refer to the papers below.

  1. Line 152 and Figure 1 indicate that only a subset of the box sizes are considered when finding the best fitting regression line (i.e., the blue line and red line have different ranges). The justification and potential implications of this choice are not clear. For instance, why not fit a regression to the full range and calculate its slope? How much heterogeneity is there in the ranges of selected box sizes between different regions? Between different participants?

Author’s response: We thank reviewer’s suggestion. Please refer to the response of comment 9. In this study, we found the range of box size from 2 to 8 mm can reach the highest slope correlation.

  1. Line 163: What values are being correlated in this approach? Presumably this is referring to FD values across individual participants, but this is not clearly stated.

Author’s response: we apologize for the misunderstanding. In the revised manuscript, we have rewritten the paragraph and the revised manuscript is as follows:

“There are two steps to build a structural brain network: one is to calculate the correlation between pairs of sub-regions to indicate the strength of the inter-regional connectivity. Thus, the brain structure network is derived from the 68*68 correlation matrix of the FD values of the paired regions. Secondly, we used a modular analysis to separate the different brain distinctions into modules based on their inter-regional connections [27].”

  1. Line 163: The definition of modularity (Q) is unclear as stated. Please be more explicit and provide the equation that was used.

Author’s response: we have added a paragraph and equation in section 2.4 to explain the definition of modularity (Q). the added paragraph is as follows:

  1. Line 170: What threshold was used to define edges and modules in the current analyses is the justification of this threshold? Are the results consistent across a range of threshold values? This is of course a very common concern for network analyses.

Author’s response: we thank reviewer’s suggestion. In the revised manuscript, we had added a paragraph to explain the concept, and the added paragraph is as follows

“In this study, we have set a proportional value of 0.2 as a threshold to filter the connectivity matrix by preserving 20% proportion of the strongest correlation coefficients. In this process, all other entries below the threshold, negative correlations and all entries on the main diagonal (self-to-self connections) are set to 0 and the links will not exist. “

  1. Line 190: “A smaller number of participation coefficient indicates a large degree of intra-modal connectivity within node i”. Is this true? Shouldn’t a smaller participation coefficient simply indicate a smaller degree of inter-modal connectivity at node i?

Author’s response: we thank reviewer’s comment. In the revised manuscript, we have rewritten the paragraph and the revised paragraph is as follows:

  1. Statistical analyses: The authors are testing for regional differences in FD in 68 ROIs, and thus increase the type 1 error rate. It is necessary to apply appropriate correction for multiple comparisons.

Author’s response: we have rewritten the paragraph, and performed FDR and Effect Size test. The revised paragraph is as follows:

“2.5. Statistical Analysis

In this study, a two-tailed t-test and multiple false discovery rate (FDR) correction were used to compare measurements [29], including FD values and brain structural network parameters, between the control and MCI patient groups. Then, the effect size process was applied to measure the strength of the relationship between compared variables of groups to indicate the practical difference [30]. Note that each group with 30 subjects had 68 FD values for each region. We computed the FD value on the basis of correlation between any two regions. As a result, a 68 × 68 correlation map was obtained for each group to build a structural network, resulting in one set of topological properties for each structural network. Accordingly, we could not directly perform any statistical comparison of the corresponding topological properties between these two structural networks.

  1. Dey, Soumen & Delampady, Mohan. (2013). False discovery rates and multiple testing. Resonance. 18. 10.1007/s12045-013-0137-9.
  2. Kelley, Ken; Preacher, Kristopher J. (2012). "On Effect Size". Psychological Methods17(2): 137–152.

  1. Line 209: For the permutation test, why were only 10 MCI participants and 10 controls reassigned? What happens to the other 10 participants in each group - are they fixed across permutations? If so, this is an inappropriate implementation. If not, I see no reason why the full sample of 20 MCI and 20 controls shouldn’t be randomly reassigned, as is done in a typical permutation / randomization test. Further 100 permutations seems a little small. I would suggest at least 1000. Finally, this paragraph seems to contain several redundant sentences.

Author’s response: in each permutation, we randomly selected 10MCI and 10 controls to perform network analysis. We performed the process 1000 times, and got 1000 network parameters to compare the difference between groups. The revised manuscript is as follows:

“In this study, a permutation test was conducted to statistically compare the differences in network properties between the two groups [30]. To test the null hypothesis, we randomly selected 10 MCI and 10 controls from each study group (30 subjects) and reassigned these 20 subjects as the randomized MCI group and randomized control group, separately. This randomized simulation and recalculation of the network properties were repeated 1000 times to compute the correlation matrix for each randomized group. The 95th percentile points of each distribution of the 1000 simulations were used as critical values in a two-sample, one-tailed t test to reject the null hypothesis, with a type I error probability of 0.05. Then, the network properties Q, P, and Z were calculated for each reassigned correlation matrix of the two groups. Following the permutation process, 1000 sets of network parameters were used in a two-sample, one-tailed t test with FDR correction to assess significant differences between the study groups.”

  1. Line 215: It is inappropriate to run a t test on bootstrapped permutations. This creates a problem such that non-meaningful differences would be considered significant simply by adding more and more permutations. Instead, the appropriate test is to generate an “empirical p value” by calculating the proportion of permutations in which an effect (i.e., difference between group 1 and group 2) as or more extreme than the observed difference. For demonstration, see, e.g.: https://rpubs.com/valentin/permutation-test-1 or https://towardsdatascience.com/how-to-use-permutation-tests-bacc79f45749

Author’s response: please refer to the author’s response of comment 17.

  1. Line 218: The authors interpret reductions in FD values to reflect “atrophy”. What is the justification for this interpretation? How do these measures compare to more established indicators of “atrophy”, namely regional volume and cortical thickness?

Author’s response: In the revised manuscript, we have revised the sub-title of section 3.1 as: 3.1. Patients with MCI exhibited significant lateralized FD changes mainly in temporal lobes regions. Previous FD studies have reported that FD are more suitable to detect cerebral atrophy. The interpretation of reductions in FD values to reflect “atrophy” can be found from our previous study. Please refer the reference below.

This study employs 3D FD analysis to quantify the degeneration of the CBWM and CBGM in MSA-C patients. Fig. 7 shows that because the FD is an index of morphological complexity (Esteban et al., 2007), a higher FD value indicates a more complex cerebellar structure, while a decrease in the cerebellar FD value may indicate a degeneration of the cerebellar structure. Since the 3D CB of MSA-C patients manifested less folding patterns and a smaller volumetric size, smaller FD values and volumetric values were anticipated.

The revised section 3.1 ia as follows:

  1. Results

3.1. Patients with MCI exhibited significant lateralized FD changes mainly in temporal lobes regions

Table 3 summarizes the FD values of each lobe in control and MCI groups. The MCI group revealed significantly lower FD values in their bilateral temporal lobes (left temporal: p<0.001, right temporal: p=0.0025) and right parietal lobe(p=0.0015).

Table 3. FD measure results of cerebral lobes in Control and MCI groups.

Lobe

Control

MCI

P value

Effect size

Frontal(L)

2.2324+0.0161

2.2319+0.0224

0.9149

0.0128

Frontal(R)

2.2393+0.0144

2.2331+0.0213

0.1858

0.1681

Temporal (L)

2.2108+0.0108

2.1908+0.02

<0.001*

0.5283

Temporal (R)

2.1985+0.0146

2.1849+0.0189

0.0025*

0.3735

Parietal(L)

2.2902+0.0183

2.2844+0.0158

0.1922

0.1672

Parietal(R)

2.2990+0.0184

2.2822+0.0209

0.0015*

0.3924

Occipital(L)

2.2146+0.0301

2.2084+0.0299

0.4262

0.1028

Occipital(R)

2.2247+0.0334

2.2159+0.024

0.2469

0.1498

  1. Results: It would be very helpful to report the regional FD values visually plotted on a brain surface. It would also be useful to compare these spatial maps to other established measures derived from the T1 scans, such as regional volume and/or cortical thickness.

Author’s response: we thank reviewer’s suggestion. Because some cortical parcellated regions such as cingulate, insula, para hippocampal and so on are located beneath the cortical surface. Sorry, hence we used a brain volume space in this study. We will take the second suggestion of compare FD measure with volume measure or cortical thickness in our future studies.

  1. Results: Please report estimates of effect size for all analyses (in addition to p values).

Author’s response: we have rewritten the statistical analysis section paragraph, and performed FDR and Effect Size test. The revised paragraph is as follows:

“2.5. Statistical Analysis

In this study, a two-tailed t-test and multiple false discovery rate (FDR) correction were used to compare measurements [29], including FD values and brain structural network parameters, between the control and MCI patient groups. Then, the effect size process was applied to measure the strength of the relationship between compared variables of groups to indicate the practical difference [30]. Note that each group with 30 subjects had 68 FD values for each region. We computed the FD value on the basis of correlation between any two regions. As a result, a 68 × 68 correlation map was obtained for each group to build a structural network, resulting in one set of topological properties for each structural network. Accordingly, we could not directly perform any statistical comparison of the corresponding topological properties between these two structural networks.

3.1. Patients with MCI exhibited significant lateralized atrophy mainly in temporal lobes regions

Table 3 summarizes the FD values of each lobe in control and MCI groups. The MCI group revealed significantly lower FD values in their bilateral temporal lobes (left temporal: p<0.001, right temporal: p=0.0025) and right parietal lobe(p=0.0015).

Table 3. FD measure results of cerebral lobes in Control and MCI groups.

Lobe

Control

MCI

P value

Effect size

Frontal(L)

2.2324+0.0161

2.2319+0.0224

0.9149

0.0128

Frontal(R)

2.2393+0.0144

2.2331+0.0213

0.1858

0.1681

Temporal (L)

2.2108+0.0108

2.1908+0.02

<0.001*

0.5283

Temporal (R)

2.1985+0.0146

2.1849+0.0189

0.0025*

0.3735

Parietal(L)

2.2902+0.0183

2.2844+0.0158

0.1922

0.1672

Parietal(R)

2.2990+0.0184

2.2822+0.0209

0.0015*

0.3924

Occipital(L)

2.2146+0.0301

2.2084+0.0299

0.4262

0.1028

Occipital(R)

2.2247+0.0334

2.2159+0.024

0.2469

0.1498

Table 4 summarizes the sub-regions with significantly decreased FD values in MCI. MCI showed 27 sub-regions with significantly decreased FD values, and mainly in the right hemisphere (L/R:10/17). Of these 27 significantly decreased FD sub-regions, 11 were in the temporal lobe (L/R:6/5), 7 in the frontal lobe (L/R:2/5), 7 in the parietal lobe(L/R:2/5) and 2 in the right occipital lobe.

Table 4. Cortical sub-regions of significant decreased FD values in MCI group.

Frontal(L)

Left hemisphere

controls

MCI

P value

Effect size

Paracentral

2.1765+0.0523

2.1546+0.0443

0.0268

0.22

Rostral middle frontal

2.4095+0.0178

2.4009+0.0247

0.0353

0.20

Frontal(R)

Caudal anterior cingulate

2.1322+0.0554

2.1037+0.0497

0.0196

0.26

Caudal middle frontal

2.2911+0.0448

2.2766+0.0335

0.0363

0.18

Medial Orbital Frontal 

2.2473+0.0359

2.2235+0.0776

0.0342

0.19

Paracentral

2.2118+0.0361

2.1724+0.0378

0.0003

0.47

Superior frontal

2.4082+0.0169

2.3988+0.0166

0.0189

0.27

Temporal(L)

Medial Temporal          

2.3148+0.0274

2.2970+0.0334

0.0245

0.28

Fusiform

2.2993+0.0271

2.2853+0.0248

0.0183

0.26

Inferior temporal

2.3358+0.020

2.3223+0.0294

0.0182

0.26

Transverse Temporal

2.0486+0.0498

2.0229+0.0595

0.0248

0.23

Entorhinal

2.1278+0.0415

2.0766+0.0518

0.0004

0.47

Temporal pole

2.1937+0.0352

2.1734+0.0445

0.0213

0.25

Temporal(R)

Bankssts   

2.1815+0.0385

2.1622+0.0562

0.0352

0.20

Fusiform    

2.2994+0.0268

2.2816+0.0341

0.0195

0.28

Para hippocampal   

2.0464+0.0590

2.0165+0.050

0.0200

0.26

Transverse Temporal    

1.9913+0.0621

1.9542+0.0654

0.0216

0.28

Superior temporal

2.3435+0.0351

2.333+0.0281

0.0438

0.16

Parietal(L)

  Postcentral

2.2919+0.02879

2.2807+0.0267

0.0368

0.20

Supra marginal

2.3867+0.0271

2.3737+0.0263

0.0242

0.28

Parietal(R)

Inferior parietal

2.4292+0.020

2.4181+0.0171

0.0275

0.27

Superior parietal

2.3581+0.0221

2.3424+0.0193

0.0076

0.35

Postcentral

2.2829+0.02169

2.2624+0.0279

0.0047

0.38

Poster cingulate

2.2097+0.0602

2.1864+0.0652

0.0362

0.18

Supra marginal

2.3886+0.02437

2.3574+0.0329

0.0342

0.47

Occipital(R)

Lateral occipital

2.3713+0.0256

2.3619+0.0261

0.0357

0.18

lingual

2.2721+0.0257

2.2528+0.0373

0.0288

0.29

     In result section 3.3

“After 1000 permutations were calculated, the network modularity value (Q) was significantly lower in the MCI group than in the control group (normal: 0.2548+0.0057, MCI: 0.2451+0.0066, P < 0.05, Effect Size=0.62).”

The effect size calculators

https://lbecker.uccs.edu/#Calculate%20d%20and%20r%20usaaing%20t%20values%20(separate%20groups)

  1. Table 3: The authors interpret the asymmetric difference in regional FD differences to reflect patterns of “lateralized atrophy”. However, the lack of symmetry / anatomical consistency might also imply that these differences are instead driven by noise, given the relatively small sample size. Moreover, the p values are relatively high and most of them are not likely to survive correction for multiple comparisons.

Author’s response: we thank reviewer’s suggestion, the original Table 3 had been revised as Table 4 in the revised manuscript and had added the effect size value please in Table 4, please refer to response for comment 20.

  1. Figure 2: To aid clarity of interpretation, please use the same color scale / range for panels A and B.

Author’s response: we greatly appreciate reviewer’s suggestion. The revised figure 2 is as follows:

       Figure 2. Sub-region correlation maps between different brain lobes in the control group (Figure 2(a)) and MCI group (Figure 2(b)). There are 68 rows and 68 columns in each plot, and the dots from the first row and first column from the top left indicate the correlation rate of the first sub-region of the ROI (left anterior cingulate) with the other 67 sub-regions, the second row and second column from the top left indicate the correlation rate of the second sub-region of the ROI (right anterior cingulate) with the other 67 sub-regions, and so on. In each figure, sub-regions of the frontal lobe are labeled within the red line, the temporal lobe is labeled within the purple line, the parietal lobe is labeled within the white line, and the occipital lobe is labeled within the green line. The color bars indicate the density of correlation, and the color bars of the control group are scaled higher than those of the MCI group. For each brain lobe, normal controls showed higher correlation densities within the lobes and with other lobes than the MCI group.

  1. What is the justification that correlating patterns of FD should be anatomically meaningful? Moreover, why would these expected to be organized according to lobes? An expected property of valid brain “networks” would be the observation of strong positive correlations within modules and weak correlations between modules. With the possible exception of the parietal lobe in controls (Figure 2A), I am not convinced that these data capture meaningful network organization.

Author’s response: In brain structure networks, regional brain thickness or volume is used as a nodal factor to calculate the correlation between nodes. Nodes with high correlation are edges and groups of nodes with high correlation are modules.  In this study, we use the intra-modular connectivity(Zi) of each sub-region within lobe as the nodes in a module to calculate the intra-module connectivity. The intra-lobe connectivity was driven from network analysis, not directly from figure 2A or figure 2B.

  1. Moreover, the correlation plot in MCIs (Figure 2B) appears to be largely driven by noise. This raises the potential limitation of data quality in MCIs. As these participants are noted to have observable atrophy, this may contribute to difficulty in segmenting and parcellating the cortical surface, which may have downstream effects on the reliability of the FD estimates. What steps were taken to ensure that the MRI data and inputs for the FD calculation (particularly in the MCI sample) were of acceptable quality?

Author’s response: The thickness of the MRI slice is very important for FD measurements. If the thickness of the slice is greater than the size of the box used to calculate the FD, then it is not possible to obtain a correct FD value. The thickness of the slice must be smaller than the size of the counting box. In this study, the minimum size of the counting box was 2 mm, while the thickness of the MRI slice was 1.5 mm. Therefore, in this study, the FD calculation was appropriate for patients with MCI and had an accurate result.

  1. Lines 243-254: The authors provide an inadequate qualitative description (more or fewer blue, green, or yellow dots) to address quantitative research questions. What were the average correlation values within and between the lobes (numeric correlation ratio and intra-lobe connectivity values)? How do those numeric values differ between MCI and controls?

Author’s response: we greatly appreciate reviewer’s suggestion. In the revised manuscript, we have rephrased section 3.2 and added the average correlation values within and between the lobes as the reviewer’s suggestion. The revised manuscript is as follows:

“ Figure 2(a) and (b) illustrate the correlation map of sub-regions between different lobes of control and MCI groups. The color bar indicates the correlation rate. Firstly, the control participants group shows a higher range of correlation rate from zero to 0.8, while the MCI group shows a lower correlation rate from zero to 0.65. For controls, the mean correlate rate of frontal lobe is 0.3276, 0.2754 for the temporal lobe, 0.3719 for the parietal lobe, and 0.2888 for the occipital lobe. The MCI revealed a lower correlation rate in each lobe. They sowed the correlate rate of 0.1679 for the frontal lobe, 0.1861 for the temporal lobe, 0.1451 for the parietal lobe, and 0.1540 for the occipital lobe.”

  1. Line 255: Patterns of correlation among FD values do not offer any validated index of (presumably anatomical) “links” or “functional dissociation between lobes”. This statement seems to conflate the current methods with other methods to measure structural and functional brain networks with DTI and FC.

Author’s response: the revised manuscript is as follows:

” Figure 2(a) and (b) illustrate the correlation map of sub-regions between different lobes of control and MCI groups. The color bar indicates the correlation rate. Firstly, the control participants group shows a higher range of correlation rate from zero to 0.8, while the MCI group shows a lower correlation rate from zero to 0.65. For controls, the mean correlate rate of frontal lobe is 0.3276, 0.2754 for the temporal lobe, 0.3719 for the parietal lobe, and 0.2888 for the occipital lobe. The MCI revealed a lower correlation rate in each lobe. They sowed the correlate rate of 0.1679 for the frontal lobe, 0.1861 for the temporal lobe, 0.1451 for the parietal lobe, and 0.1540 for the occipital lobe.”

2.4. Network property analysis of intra-modular and inter-modular connectivity

There are two steps to build a structural brain network: one is to calculate the correlation between pairs of sub-regions to indicate the strength of the inter-regional connectivity. Thus, the brain structure network is derived from the 68*68 correlation matrix of the FD values of the paired regions. Secondly, we used a modular analysis to separate the different brain distinctions into modules based on their inter-regional connections [27].  Modularity (Q) indicates the number of edges where all pairs of nodes within the same module.

The Q can be expressed as:

where A is the connection matrix of the network, and each element of A is the correlation coefficient between regions;  is defined as the sum of the correlation coefficient between node i and its connected regions, and is also called the degree of node I;  represents the total number of edges; and  denotes the module of node i. The  -function  is 1 when nodes i and j belong to the same module and 0 otherwise. If the investigated network presents superior partitioning, it will have a greater Q value and is more likely to construct a modular organization [21].

In this study, we have set a proportional value of 0.2 as a threshold to filter the connectivity matrix by preserving 20% proportion of the strongest correlation coefficients. In this process, all other entries below the threshold, negative correlations and all entries on the main diagonal (self-to-self connections) are set to 0 and the links will not exist.”

  1. Line 257: If this test was performed using the t test procedure described above, it is inappropriate. Instead, the authors should calculate the difference in Q values between groups for each of the permutations (ideally at least 1000). Then, the p value can be calculated as the proportion of permutations in which a difference as or more extreme than the observed difference was detected (see comments and links above).

Author’s response: In the revised manuscript, we have performed 1000 times of permutation and have 1000 Q values for MCI and control groups. We compare the 1000 Q values between MCI and controls with FDR correction and take the mean of the 1000 p values.

“2.5. Statistical Analysis

In this study, a two-tailed t-test and multiple false discovery rate (FDR) correction were used to compare measurements [29], including FD values and brain structural network parameters, between the control and MCI patient groups. Then, the effect size process was applied to measure the strength of the relationship between compared variables of groups to indicate the practical difference [30]. Note that each group with 30 subjects had 68 FD values for each region. We computed the FD value on the basis of correlation between any two regions. As a result, a 68 × 68 correlation map was obtained for each group to build a structural network, resulting in one set of topological properties for each structural network. Accordingly, we could not directly perform any statistical comparison of the corresponding topological properties between these two structural networks.

In this study, a permutation test was conducted to statistically compare the differences in network properties between the two groups [31]. To test the null hypothesis, we randomly selected 10 MCI and 10 controls from each study group (30 subjects) and reassigned these 20 subjects as the randomized MCI group and randomized control group, separately. This randomized simulation and recalculation of the network properties were repeated 1000 times to compute the correlation matrix for each randomized group. The 95th percentile points of each distribution of the 1000 simulations were used as critical values in a two-sample, one-tailed t test to reject the null hypothesis, with a type I error probability of 0.05. Then, the network properties Q, P, and Z were calculated for each reassigned correlation matrix of the two groups. Following the permutation process, 1000 sets of network parameters were used in a two-sample, one-tailed t test with FDR correction to assess significant differences between the study groups. “

  1. 2Figure 3: The glass brain figures are somewhat unclear to interpret. Since the data was extracted from a surface-level parcellation, information regarding module assignment would be communicated more effectively by plotting the module assignment on a brain surface (e.g., using ggseg, workbench, or any preferred means of visualizing quantitative brain data on the surface rather than volume space).

Author’s response: we greatly thank reviewer’s suggestion. Because some cortical parcellated regions such as cingulate, insula, para hippocampal and so on are located beneath the cortical surface. Sorry, hence we used a brain volume space to illustrate the distribution nodes of modules.

  1. Line 281: Again, it does not seem appropriate or valid to describe differences in FD-based network analyses to reflect “functional integration”.

Author’s response: we have deleted the sentence and revised the paragraph as follows:

“3.2. Patients with MCI exhibited lower correlation rates within and between lobes

Figures 2(a) and (b) illustrate the correlation plots between different brain lobes in the control and MCI groups for the sub-regions. The color bars indicate the correlation rates. First, the control group shows a higher correlation rate range from 0 to 0.8, while the MCI group shows a lower correlation rate from 0 to 0.65. For the control group, the mean correlation rate is 0.3276 for the frontal lobe, 0.2754 for the temporal lobe, 0.3719 for the parietal lobe, and 0.2888 for the occipital lobe. MCI shows lower correlation rates for each lobe. They showed correlation rates of 0.1679 for the frontal lobe, 0.1861 for the temporal lobe, 0.1451 for the parietal lobe, and 0.1540 for the occipital lobe.”

  1. Section 3.3: What methods / sources were used to assign functional associations to ROIs?

Author’s response: In the revised version, we removed the sentences associated with these functions from the original draft. We have rewritten and shortened the Result section 3.3, and the revised section 3.3 reads as follows.

“3.3. MCI revealed smaller modular size and less nodes integration in their brain structural network

Figures 3(a) and (b) illustrate the node distribution of modules in the FD-based brain structure network for the control and MCI groups, respectively. The figures were plotted using BrainNet Viewer software [32]. According to the network analysis, the 68 sub-regions of the cortex in the control group were clustered into five middle segments, whereas the 68 sub-regions of the cortex in MCI were clustered into six modules.

Table 5(a) and Table 5(b) summarize the detailed sub-regions within each module of the brain module network for the normal and MCI groups, respectively. In each module, the top row indicates nodes in the left hemisphere and the bottom row indicates nodes in the right hemisphere. In this study, we defined the largest module as the first module, the second largest module as the second module, and so on. The control group had three larger modules (Module 1, 19 nodes; Module 2, 17 nodes; Module 3, 17 nodes), and two smaller modules (Module 4, 8 nodes; Module 5, 7 nodes). The MCI showed smaller module sizes in their brain network than the control group. There was only one larger module (module 1, 18 nodes), three medium-sized modules (module 2, 13 nodes; module 3, 11 nodes; module 4, 10 nodes), and two smaller modules (module 5, 9 nodes; module 6, 7 nodes) in MCI's brain network. After 1000 permutations were calculated, the network modularity value (Q) was significantly lower in the MCI group than in the control group (normal: 2548, MCI: 0.2451, P < 0.05, FDR corrected). This result implies a relatively low density and efficiency of the structural brain network in MCI.

For the control group, modules 1, 2 and 3 all show the integration of nodes from the four functional lobes (frontal, temporal, parietal and occipital). In module 1, there were three pairs of bilateral nodes and in module 2, there were four pairs of bilateral nodes (bolded in Table 5(a)). Unlike controls, MCI showed sparse clustering and less functional node integration in its brain structural network module groupings. In MCI's modules 1, 2, and 3, nodes in the frontal, parietal, and temporal lobes were integrated, but none of the nodes in the occipital lobe. MCI also showed fewer bilateral node pairs in its brain structural network modules. In Module 1, there are two bilateral node pairs linked. In modules 2 and 3 of MCI, there are only one pair of bilateral node links (bolded in Table 5(b)).”

  1. Section 3.3: How do the identified modules correspond to more established brain networks, e.g., default mode, fronto-parietal, etc.?

Author’s response: We have added a paragraph to the Discussion section to explain the correspondence of our results with other functional network studies, and the added paragraph in the discussion section is as follows

“4.2. MCI show shrinkage modular size and less functional lobes integration in their brain structural network

 Structural networks are believed to shape and provide constraints for the dynamics of functional connectivity, to some extent, it has been widely acknowledged that functional networks can be predicted from the underlying structural connectome [39]. A high goodness of fit level for the structure-function mapping of brain networks has been reported [40,41], as well as a pattern dependence between the connection matrices of the resting-state functional and structural networks [41]. Robust model analysis also reflects a reliable combination of structural and functional networks that are optimally correlated, with the structural network predicting the functional network, but the two networks not necessarily overlapping [40].

Functional network studies have reported that cerebral sub-regions that exhibit different combinations of control signals in many tasks can be grouped into three distinct networks, namely the Fronto-Parietal network (FPN), Cingulo-Opercular network (CON) and default model network (DMN) [42-44]. The FPN includes the prefrontal, middle cingulate gyrus, inferior parietal and precuneus, the CON includes the prefrontal, insula, anterior cingulate and superior frontal lobes, and the DMN includes the inferior temporal, para hippocampal, lateral parietal and posterior cingulate gyrus [42].

In the present study, we investigated and compared the brain structural network patterns in normal controls and MCI group. In the control group, we found network model 1 included many of the nodes of the FPN (frontal, parietal and precuneus). Network model 2 included many of the nodes of the DMN (para hippocampal (Left and right) and posterior cingulate (left and right)). Network model 3 included many nodes of the CON (anterior cingulate, superior frontal and insula).”

The added references are as follow:

  1. O’Neill, G.C., Tewarie, P., Vidaurre, D., Liuzzi, L., Woolrich, M.W., Brookes, M.J., 2017. Dynamics of large-scale electrophysiological networks: a technical review. Neuroimage 180, part B, 559–576.
  2. Miši´c, B.; Betzel, R.F.; De Reus, M.A.; Van Den Heuvel, M.P.; Berman, M.G.; McIntosh, A.R.; Sporns, O. Network-level structure-function relationships in human neocortex. Cereb. Cortex 2016, 26, 3285–3296.
  3. Meier J, Tewarie P, Hillebrand A, Douw L, van Dijk BW, Stufflebeam SM, Van Mieghem P. A Mapping Between Structural and Functional Brain Networks. Brain Connect. 2016 May;6(4):298-311. doi: 10.1089/brain.2015.0408. Epub 2016 Mar 29. PMID: 26860437; PMCID: PMC4939447.
  4. Fair DA, Cohen AL, Power JD, Dosenbach NUF, Church JA, et al. (2009) Functional Brain Networks Develop from a ‘‘Local to Distributed’’ Organization. PLoS Comput Biol 5(5): e1000381. doi:10.1371/journal.pcbi.1000381
  5. Dosenbach NU, Fair DA, Miezin FM, Cohen AL, Wenger KK, et al. (2007) Distinct brain networks for adaptive and stable task control in humans. Proc Natl Acad Sci U S A 104: 11073–11078.
  6. Dosenbach NU, Visscher KM, Palmer ED, Miezin FM, Wenger KK, et al. (2006) A core system for the implementation of task sets. Neuron 50: 799–812.
  7. Line 346: The authors cannot infer that patterns of correlation in FD values indicate “association fibers”.

 Author’s response: Sorry for the misuse error. In the revised manuscript, we have deleted all the words “association fiber”. In the revised manuscript, we have rephrased and shortened the results section. In the original manuscript, line 346 was in results section 3.4. The revised section 3.4 is as follows

“3.4. MCI revealed significant alteration of intra-lobe and inter-lobes connectivity in their brain structural network

Figures 4(a) to (h) illustrate the detailed connections within each brain lobe in the control and MCI groups, and Table 6 summarizes the number of connections within the left hemisphere, left hemisphere and between hemispheres. First, the link distribution in each lobe of MCI showed a similar pattern to that of the control group. In each brain lobe, MCI showed a smaller number of links and thinner link widths. The MCI group showed the most lateral link loss in the left temporal lobe (controls: 9 links, MCI: 4 links) and also the most bilateral link loss in their temporal lobe (controls: 24 links, MCI: 14 links). Compared with Figure 4(c) and (g), only two red lines in Figure 4(g) are wider than those in Figure 4(c), which may imply that the MCI group has the most severe decrease in bilateral link strength in the parietal lobe.”

  1. Line 352: What test was performed to determine that “MCIs showed significantly decreased intra-lobe connectivity in all cerebral lobes”?

Author’s response: we thank reviewer’s suggestion. In the revised manuscript, we have added a table as Table 5 to indicate the connection number within lateral lobe and between bilateral lobes. We also have rearranged and relabeled the right (R) and left (L) lobes in Figure 4 to make it more easily to read. The revised manuscript is as follows:

3.4. MCI revealed significant alteration of intra-lobe and inter-lobes connectivity in their brain structural network

Figures 4(a) to (h) illustrate the detailed connections within each brain lobe in the control and MCI groups, and Table 6 summarizes the number of connections within the left hemisphere, left hemisphere and between hemispheres. First, the link distribution in each lobe of MCI showed a similar pattern to that of the control group. In each brain lobe, MCI showed a smaller number of links and thinner link widths. The MCI group showed the most lateral link loss in the left temporal lobe (controls: 9 links, MCI: 4 links) and also the most bilateral link loss in their temporal lobe (controls: 24 links, MCI: 14 links). Compared with Figure 4(c) and (g), only two red lines in Figure 4(g) are wider than those in Figure 4(c), which may imply that the MCI group has the most severe decrease in bilateral link strength in the parietal lobe.

  1. Figure 4: The presentation of left vs. right vs. interhemispheric correlations using different colors is somewhat unclear. It may be simpler to plot the left hemisphere ROIs on the left side of the circle and the right hemispheric ROIs on the right, as was done in Figure 5. Then hemispheric information can be easily interpreted among the correlations.

Author’s response: please refer to the author’s response of comment 33.

  1. Much of the text throughout the results is redundant, repetitive, and simply describes the figures without providing necessary quantitative analysis. For instance, descriptions of the color labeling of points should be put in the figure captions, not the text.

Author’s response: We have rewritten and shorten the result section. We have added more quantitative analysis in the revised manuscript. We also have replaced the descriptions of the color labeling of points in the figure captions. Please refer to the revised manuscript.

  1. Line 387: Again “significance” is determined by visually examining a figure without reporting results of an appropriate statistical test.

Author’s response: We have deleted these sentences in the revised manuscript. We also we improve more appropriate statistical test as the reviewer had suggested, and have improved the original manuscript more organized and easy to read, thanks!

  1. Discussion: The authors are again over-interpreting their results. I see no justification that patterns of correlation between FD may reflect “commissural fibers”, “long association fibers”, “new wiring”, or “compensation”.

Author’s response: We have deleted all these words “commissural fibers”, “long association fibers”, “new wiring”, or “compensation” in the revised manuscript.

  1. Line 439: The authors did not compare FD to any established measure of atrophy (i.e., cortical thickness or volume) for detecting differences in MCI. They have not justification to claim that FD is “superior” in any way or that it is even a measure of “atrophy.”

Author’s response: we thank reviewer’s suggestion. In the revised manuscript, we have added a paragraph to compare our results with other measure of atrophy in MCI. The added paragraph in the revised discussion section 4.1 is as follows:

“4.1. FD measure revealed ability for detecting of cerebral changes in MCI

The FD approach is a consistent and most frequently chosen feature that has been proposed to calculate the intrinsic structural complexity of the cerebral cortex to predict cognitive decline in disease and can complement standard imaging [33]. Traditional methods such as cortical thickness or volume show that MCI may atrophy their cerebral cortex mainly in the medial temporal, hippocampus, entorhinal, and some sporadic reports in the para hippocampus, amygdala, fusiform gyrus, lateral temporal, parietal, frontal and occipital lobes [6-8]. However, in neurodegenerative diseases, the complexity of assessing cortical shape may better reflect symptoms of atrophy than using traditional volumetric measures [34]. In this study, we prospectively applied FD to measure cortical DK sub-regions in MCI, and the regions of atrophy we identified included those measured by conventional methods in previous studies [6-8], as well as additional sub-regions in the medial orbital frontal, paracentral, inferior parietal, and superior parietal lobes. Our results showed that the medial temporal, para hippocampal, paracentral, entorhinal, fusiform, postcentral and superior parietal were the sub-regions with more decreased FD values in MCI.

Using the same FD analysis, Nicolas Nicastro et al. reported that the orbitofrontal cortex and paracentral gyrus are particularly vulnerable in terms of memory and language impairment, and that the FD represents a sensitive imaging marker for prevention and diagnostic strategies [34]. In subjects with MCI, precise measurement of medial temporal lobe atrophy (MTA) may improve predictive accuracy and reduce false-negative classification of dementia [35]. Furthermore, it has been highlighted that visual assessment of MTA on brain MRI using a standardized rating scale is a strong and independent predictor of conversion to dementia in relatively young MCI patients [36]. With increasing duration of MCI, measuring hippocampal atrophy in older MCI patients has been reported to predict subsequent conversion to AD [37]. Structural abnormalities in the orbitofrontal cortex (OFC) may reflect a potential neurodevelopmental risk marker for MCI [38]. Taken together, our results support these previous findings in MCI and may provide a new approach for identifying MCI.”

Round 2

Reviewer 1 Report

The authors have updated the work according to the comments made.